# WHERE MLLMs ATTEND AND WHAT THEY RELY ON: EXPLAINING AUTOREGRESSIVE TOKEN GENERATION

## ABSTRACT

Multimodal large language models (MLLMs) have demonstrated remarkable capabilities in aligning visual inputs with natural language outputs. Yet, the extent to which generated tokens depend on visual modalities remains poorly understood, limiting interpretability and reliability. In this work, we present EAGLE, a lightweight black-box framework for explaining autoregressive token generation in MLLMs. EAGLE attributes any selected tokens to compact perceptual regions while quantifying the relative influence of language priors and perceptual evidence. The framework introduces an objective function that unifies sufficiency (insight score) and indispensability (necessity score), optimized via greedy search over sparsified image regions for faithful and efficient attribution. Beyond spatial attribution, EAGLE performs modality-aware analysis that disentangles what tokens rely on, providing fine-grained interpretability of model decisions. Extensive experiments across open-source MLLMs show that EAGLE consistently outperforms existing methods in faithfulness, localization, and hallucination diagnosis, while requiring substantially less GPU memory. These results highlight its effectiveness and practicality for advancing the interpretability of MLLMs.

## 1 INTRODUCTION

Multimodal large language models (MLLMs) (Achiam et al., 2023; Wang et al., 2025; Bai et al., 2025; Comanici et al., 2025) have achieved significant progress in vision–language understanding and generation. By jointly modeling visual and textual modalities, they can now perform a wide range of tasks, such as image captioning and visual question answering (VQA) (Li et al., 2025b). These advances have enabled MLLMs to approach human-level performance on many benchmarks and to underpin various real-world applications (Liang et al., 2024; Li et al., 2024). However, alongside these advances come critical challenges in transparency and reliability (Zhang et al., 2025b). As parameter scales and modality coverage continue to expand, MLLMs become increasingly opaque, making it difficult to trace how specific inputs influence generated outputs (Xing et al., 2025; Chen et al., 2025c;b). Furthermore, MLLMs are susceptible to hallucinations (Chen et al., 2025b;a), which undermine trust in safety-critical domains such as healthcare (Ahmed et al., 2025) and autonomous driving (Chen et al., 2024a). These limitations highlight the urgent need for efficient and faithful attribution methods to improve decision transparency, diagnose errors, and enhance the safety and trustworthiness of MLLMs (Lin et al., 2025; Dang et al., 2024).

Attribution in MLLMs is particularly challenging because they generate tokens autoregressively, making classification-based attribution methods difficult to adapt. Attention visualization approaches (Ben Melech Stan et al., 2024) often fail to capture complex cross-modal interactions, while gradient-based extensions (Zhang et al., 2025b; Xing et al., 2025) aggregate token logits but remain confounded by textual priors. More recently, TAM (Li et al., 2025a) employed activation maps to explain individual tokens and showed promising localization on Qwen2-VL (Wang et al., 2024), yet it cannot generalize to all MLLMs or capture multi-token contributions. In summary, attribution methods based on activation maps or gradients face inherent limitations: (1) activation-based approaches lack a direct causal link between inputs and outputs, reflecting only intermediate layer preferences often misaligned with human intuition; and (2) gradient-based approaches are sensitive to cumulative effects in long sequences and easily disturbed by noise and modality imbalance.

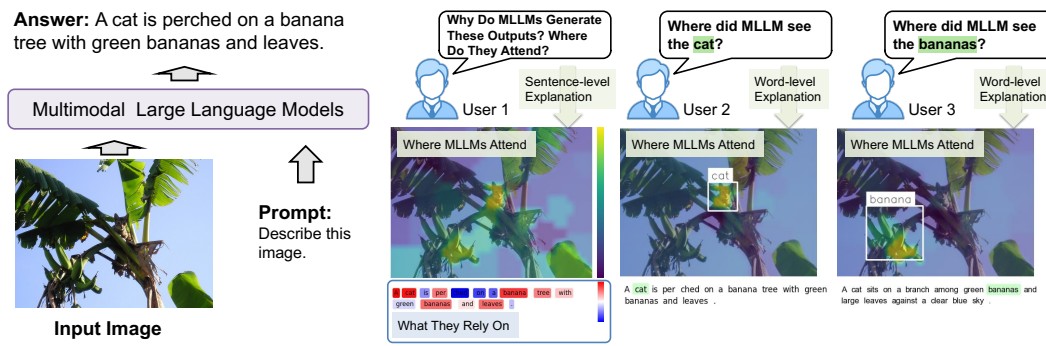

Figure 1: EAGLE attribution which perceptual regions drive the generation (Where MLLMs Attend) and quantifies modality reliance (What They Rely On).

To more faithfully explain the generation of MLLMs, we propose EAGLE (**E**xplaining **A**utoregressive **G**eneration by **L**anguage priors or **E**vidence), a black-box attribution framework for interpreting autoregressive token generation. As shown in Fig 1, our method supports attribution for any chosen set of output tokens, revealing the perceptual regions that drive their generation and quantifying the relative roles of language priors and visual evidence. Inspired by submodular subset selection, we aim to find the minimal set of perceptual regions that maximizes token logits, conditioned on the prompt and context. We design an objective function with two components: the insight score, capturing regions sufficient for generation, and the necessity score, identifying regions whose removal impairs generation. By applying greedy search over sparsified image regions, we construct an ordered ranking that attributes which perceptual regions promote generation in MLLMs, addressing the question of "**Where MLLMs Attend**". Beyond spatial attribution, we also assess "**What They Rely On**". By tracking how token logits evolve as salient regions are progressively introduced, we measure whether each token depends more on perceptual evidence or language priors, offering a faithful and comprehensive view of model decisions.

We evaluate our method on open-source MLLMs, including LLaVA-1.5 (Liu et al., 2024), Qwen2.5-VL (Bai et al., 2025), and InternVL3.5 (Wang et al., 2025), using the MS COCO Lin et al. (2014) and MMVP (Tong et al., 2024) datasets for image captioning and VQA. On faithfulness metrics, our approach outperforms existing attribution methods (LLaVA-CAM (Zhang et al., 2025b), IGOS++ (Xing et al., 2025), and TAM (Li et al., 2025a)) by an average of 20.0% in insertion and 13.4% in deletion for image captioning, and by 20.6% and 8.1% on the same metrics for VQA. At the word level, our method achieves more rational explanations of object tokens, surpassing TAM by 36.42% and 42.63% on the Pointing Game under box-level and mask-level annotations, respectively. Finally, on the RePOPE benchmark Neuhaus & Hein (2025) for object hallucination, our method accurately localizes the visual elements responsible for hallucinations and mitigates them by removing only a minimal set of interfering regions. These results demonstrate the versatility of our method across diverse tasks and benchmarks.

In summary, the contributions of this paper are:

1. We propose EAGLE, a lightweight black-box attribution framework for autoregressive token generation, which attributes any selected set of tokens to compact perceptual regions with low GPU memory cost.

2. An objective function that unifies sufficiency (insight score) and indispensability (necessity score), optimized via a greedy search strategy that balances interpretability with efficiency, yielding faithful attributions.

3. A modality analysis that quantifies whether each generated token is driven more by language priors or perceptual evidence, enabling finer-grained interpretability.

4. Experiments across diverse MLLMs show state-of-the-art interpretability in faithfulness, localization, and hallucination diagnosis.

## 2 RELATED WORK

**Multimodal LLMs Attribution.** Research on input-level attribution for Multimodal Large Language Models (MLLMs) is still nascent. LVLM-Interpret (Ben Melech Stan et al., 2024) visualizes alignment between LLaVA outputs and images using raw attention, while LLaVA-CAM (Zhang et al., 2025b) adapts Smooth-CAM (Omeiza et al., 2019) to token-level probabilities, but both suffer from layer

Figure 2: Overview of the proposed EAGLE framework. The input image is first sparsified into sub-regions, then attributed via greedy search with the designed objective, and finally analyzed for modality relevance between language priors and perceptual evidence.

sensitivity and limited faithfulness. VPS (Chen et al., 2025b) introduces a search-based method for object-level tasks, yet it is restricted to grounding and detection. IGOS++(Xing et al., 2025) identifies visually aligned tokens but remains parameter-sensitive. More recently, TAM(Li et al., 2025a) reduces contextual noise in activation maps, improving token-level attribution. However, gradient-based methods remain memory-intensive and unstable. In contrast, we propose a black-box attribution framework that localizes outputs to compact input regions without relying on token selection, quantifies the influence of language priors versus perceptual evidence, and further explains the causes of object hallucinations in MLLMs.

**Interpreting Hallucinations in MLLMs.** Several studies have applied interpretability techniques to examine hallucinations. Jiang et al. (2025) investigated how image latent representations in vision-language models are projected into the language vocabulary, thereby shaping the model's confidence in both "real" and "hallucinatory" objects, and further proposed a representation correction method to mitigate hallucinations. Zhang et al. (2025a) examined whether MLLMs attend to incorrect regions when producing wrong answers, leveraging their internal attention maps. VaLSe (Chen et al., 2025a) employs gradient- and attention-based attribution maps to identify noisy regions that contribute to hallucinations. In this work, we primarily focus on interpreting which input regions lead to incorrect decisions, aiming to suppress hallucinations by removing as few regions as possible.

## 3 METHOD

### 3.1 TASK FORMULATION

For a multimodal large language model (MLLM), such as a VLLM, given an input image $\mathbf{x}$ and a textual prompt, the model generates an output sequence $\mathbf{y} = [y_1, y_2, \ldots, y_l]$. Let $p(\cdot)$ denote the conditional probability distribution over the token vocabulary. The probability of generating each token is expressed as $p(y_t \mid \mathbf{x}, \texttt{Prompt}, \mathbf{y}_{<t})$, where $\mathbf{y}_{<t} = [y_1, \ldots, y_{t-1}]$ denotes the previously generated tokens.

For interpretability analysis, our objective is to identify the image regions $\mathbf{x}$ that most strongly drive the model's decisions. Image features in MLLMs are typically high-dimensional and information-dense but also redundant and less directly interpretable than text. We therefore focus on decomposing $\mathbf{x}$ into semantically meaningful subregions. Specifically, the image is sparsified into $V = \{\mathbf{x}_1, \mathbf{x}_2, \ldots, \mathbf{x}_N\}$ using the SLICO (Achanta et al., 2012) superpixel segmentation method, where $\mathbf{x}_i$ denotes the $i$-th subregion. The attribution problem is then cast as a subset selection task (Chen et al., 2024b): $\max_{S \subseteq V, |S| < k} \mathcal{F}(S)$, where $k$ is the maximum number of selected subregions and $\mathcal{F}(\cdot)$ is a set function measuring interpretability. Beyond the unordered case, attribution also depends on the order in which regions contribute to the decision. We therefore extend the formulation to ordered subsets:

$$\max_{\pi \in \mathcal{P}(V), |\pi| < k} \sum_{r=1}^{|\pi|} \mathcal{F}(\pi_{:r}), \qquad (1)$$

where $\pi$ is an ordered subset, $\mathcal{P}(V)$ the collection of all ordered subsets of $V$, and $r$ the prefix length. The problem thus reduces to designing $\mathcal{F}(\cdot)$ and optimizing it efficiently.

## 3.2 EXPLAINING AUTOREGRESSIVE GENERATION

We propose EAGLE, a novel attribution framework for explaining autoregressive token generation, as shown in Fig. 2. For the set function in Eq. 1, we design a submodular-inspired objective to measure interpretability. This objective encourages diminishing returns as more regions are added, although it may not be strictly submodular for MLLMs. Let $T = [t_1, t_2, \ldots, t_n]$ denote the token positions of interest, and $\mathcal{V} = [v_1, v_2, \ldots, v_n]$ their corresponding vocabulary indices.

**Insight Score:** A key metric for interpretability is the identification of the minimal set of input regions sufficient to maximize the probability of generating the target label, thereby highlighting the most informative evidence underlying the model's decision. Given an input prompt and an image $\mathbf{x}$, we denote the corresponding target sequence as $\mathbf{y}$, which is generated conditioned on both. For a candidate subregion $S$, the insight score is defined as:

$$s_{\text{insight}}(S, \texttt{Prompt}, \mathbf{y}, T, \mathcal{V}) = \sum_{i=1}^{n} p(y_{t_i} = v_i \mid S, \texttt{Prompt}, \mathbf{y}_{<t_i}), \tag{2}$$

where $p(y_{t_i} = v_i \mid S, \texttt{Prompt}, \mathbf{y}_{<t_i})$ denotes the probability of generating the ground-truth token $y_{t_i}$ at position $t_i$, conditioned on the selected subregion $S$, the input prompt, and the previously generated tokens.

**Necessity Score:** Another key metric for interpretability is the identification of the minimal set of input regions whose removal leads to a significant decrease in the probability of generating the target label, thereby revealing the indispensable evidence that the model relies on. Formally, for a candidate subregion $S$, the necessity score is defined as:

$$s_{\text{necessity}}(V \setminus S, \texttt{Prompt}, \mathbf{y}, T, \mathcal{V}) = \sum_{i=1}^{n} \left( 1 - p(y_{t_i} = v_i \mid V \setminus S, \texttt{Prompt}, \mathbf{y}_{<t_i}) \right), \tag{3}$$

where $V \setminus S$ denotes the remaining regions after removing $S$. This score provides an effective criterion in the search phase for uncovering subtle but critical regions that contribute to the final decision.

**Objective Function:** We integrate the insight and necessity scores into a unified objective function that jointly captures sufficiency and necessity for interpreting autoregressive token generation:

$$\mathcal{F}(S, V, \texttt{Prompt}, \mathbf{y}, T, \mathcal{V}) = s_{\text{insight}}(S, \texttt{Prompt}, \mathbf{y}, T, \mathcal{V}) + s_{\text{necessity}}(V \setminus S, \texttt{Prompt}, \mathbf{y}, T, \mathcal{V}), \tag{4}$$

where a larger objective value indicates that the selected input combination $S$ is more important and thus provides stronger interpretability.

**Saliency Map Generation:** To optimize the objective in Eq. 1, an $\mathcal{NP}$-hard problem, we adopt a greedy search strategy. At each step, the region yielding the largest marginal gain with respect to the objective function is added to the current set until the budget $k$ is reached, producing an ordered set $\pi$. Beyond ranking, it is also important to assess the relative saliency differences among subregions. We evaluate these differences by examining the marginal gains of the objective function as the ordered subset expands. A larger gain indicates that the newly added subregion remains highly influential, whereas diminishing gains approaching zero suggest that subsequent subregions contribute negligibly and exhibit limited saliency distinction. The attribution score $\mathcal{A}_i$ for a subregion $\pi_i$ within the ordered set $\pi$ is defined as:

$$\mathcal{A}_i = \begin{cases} 0 & \text{if } i = 1, \\ \mathcal{A}_{i-1} - \left| \mathcal{F}(\pi_{:i}) - \mathcal{F}(\pi_{:i-1}) \right| & \text{if } i > 1, \end{cases} \tag{5}$$

where $\pi_{:i}$ denotes the combination of the top $i$ subregions, and the attribution scores start from zero, decrease progressively with each step, and are subsequently normalized.

## 3.3 LANGUAGE PRIOR VS. PERCEPTION EVIDENCE

Beyond identifying which perceptual regions promote the generation of specific autoregressive tokens, we further analyze whether each generated token is more strongly influenced by language priors or by perceptual evidence. Existing approaches often assess token relevance to the visual modality by observing changes in probability when the input image is masked Xing et al. (2025). However,

simply comparing the probability with the full image against that without the image is not a reliable indicator of visual relevance, as the probability may first increase and then decrease when visual inputs are progressively inserted (Chen et al., 2024b). By contrast, if a token is truly irrelevant to the visual modality, its probability should remain stable regardless of how the image is modified.

To address this limitation, we leverage the ordered subset $\pi$ obtained in Section 3.2 and examine how each token is affected as the subregions in $\pi$ are progressively expanded, thereby quantifying the extent to which the token is influenced by perceptual evidence. Specifically, for each target token position $t_i \in T$, the influence score is defined as:

$$I_{t_i} = \sum_{r=1}^{|\pi|} \Big( p(y_{t_i} = v_i \mid \pi_{:r}, \texttt{Prompt}, \mathbf{y}_{<t_i}) - \min_{1 \leq j \leq |\pi|} p(y_{t_i} = v_i \mid \pi_{:j}, \texttt{Prompt}, \mathbf{y}_{<t_i}) \Big), \quad (6)$$

where $v_i$ denotes the vocabulary index of the target token $y_{t_i}$. The influence score $I_{t_i}$ measures the impact of perceptual evidence on the generation of token $y_{t_i}$. A larger score indicates that the token generation is more strongly driven by perceptual evidence, whereas a smaller score suggests a greater reliance on language priors, as shown in Fig. 2. The detailed calculation process of the proposed EAGLE algorithm is outlined in Algorithm 1.

**Remark 1** (Weak Submodularity). Our objective function $\mathcal{F}(\cdot)$ is not strictly submodular in MLLMs. However, it exhibits *weak submodularity*, a relaxed condition that bounds the deviation from true submodularity. Formally, let $\gamma \in (0, 1]$ denote the submodularity ratio of $\mathcal{F}$:

$$\gamma = \min_{L \subseteq U, S \subseteq U \setminus L} \frac{\sum_{i \in S} \big( \mathcal{F}(L \cup \{i\}) - \mathcal{F}(L) \big)}{\mathcal{F}(L \cup S) - \mathcal{F}(L)}.$$

When $\gamma = 1$, $\mathcal{F}$ is strictly submodular; smaller $\gamma$ values indicate weaker submodularity. Under weak submodularity, greedy selection is still guaranteed to achieve a $(1 - e^{-\gamma})$-approximation of the optimal solution (Bian et al., 2017). Thus, the stronger the submodular property of $\mathcal{F}$ in MLLMs (i.e., larger $\gamma$), the tighter the theoretical bound and the more reliable the approximation.

**Remark 2** (Token-Agnostic Attribution). Gradient-based methods (Xing et al., 2025) rely on selecting visually relevant tokens; choosing tokens dominated by language priors can distort attribution and yield unreliable explanations. In contrast, our approach is token-agnostic: even when applied to tokens strongly influenced by language priors, the visual attribution remains unaffected. Moreover, after attribution, our framework explicitly evaluates whether the selected tokens are primarily driven by perceptual evidence or language priors.

**Remark 3** (Interactive Token-Level Explanation). Our framework also allows users to select specific sentences, words, or tokens for targeted attribution. This flexibility enables fine-grained interpretation at arbitrary granularity and naturally supports human-in-the-loop analysis and interactive explanation.

**Remark 4** (Computational Complexity). The algorithm has time complexity $\mathcal{O}(2^{|V|})$. With the greedy strategy, all subregions are ordered with a total of $\frac{1}{2}|V|^2 + \frac{1}{2}|V|$ inferences, yielding a time complexity of $\mathcal{O}(|V|^2)$. The space complexity is $\mathcal{O}(|V|)$, only the ordered subset needs to be stored.

## 4 EXPERIMENTS

### 4.1 EXPERIMENTAL SETUP

**Datasets.** We evaluate across three representative tasks: MS COCO Caption (Lin et al., 2014; Chen et al., 2015) for image captioning, MMVP (Tong et al., 2024) for visual question answering (VQA), and RePOPE (Neuhaus & Hein, 2025) for object hallucination assessment.

**Baselines.** We compare EAGLE against state-of-the-art attribution methods for MLLMs, including gradient-based approaches (LLaVA-CAM (Zhang et al., 2025b) and IGOS++ adaptation (Xing et al., 2025)) and the activation-based method TAM (Li et al., 2025a). Note that TAM is restricted to attributing a single token at a time and cannot handle token combinations.

**Models.** We validate our approach on three multimodal large language models: LLaVA-1.5-7B (Liu et al., 2024), Qwen2.5-VL (3B and 7B) (Bai et al., 2025), and InternVL 3.5-4B (Wang et al., 2025).

**Evaluation Metrics.** We consider three categories of attribution metrics: *faithfulness*, *localization*, and *correction-oriented*. (1) Faithfulness metrics evaluate whether explanations align with the model's

Table 1: Evaluation of sentence-level faithfulness metrics (Deletion, Insertion AUC, and Average Highest Score) on the MS COCO and MMVP datasets using LLaVA-1.5, Qwen2.5-VL, and InternVL3.5.

| Datasets | MLLMs | Methods | Sentence-level Faithfulness | | | Sensitive Tokens-level Faithfulness | | | GPU Memory (↓) |
|---|---|---|---|---|---|---|---|---|---|
| | | | Ins. (↑) | Del. (↓) | Ave. high. score (↑) | Ins. (↑) | Del. (↓) | Ave. high. score (↑) | |
| MS COCO (Lin et al., 2014) (Image caption task) | LLaVA-1.5 7B (Liu et al., 2024) | LLaVA-CAM (Zhang et al., 2025b) | 0.5298 | 0.5317 | 0.6031 | 0.4124 | 0.4115 | 0.5783 | 37.25 GB |
| | | iGOS++ (w/ GNC) (Xing et al., 2025) | 0.5293 | 0.5168 | 0.6004 | 0.4101 | 0.3815 | 0.5731 | 48.18 GB |
| | | EAGLE | **0.5970** | **0.4554** | **0.6259** | **0.5344** | **0.2809** | **0.5993** | **16.07 GB** |
| | Qwen2.5-VL 3B (Bai et al., 2025) | LLaVA-CAM (Zhang et al., 2025b) | 0.4978 | 0.5562 | 0.6662 | 0.3541 | 0.4497 | 0.6424 | 28.99 GB |
| | | iGOS++ (w/ GNC) (Xing et al., 2025) | 0.5328 | 0.4891 | 0.6672 | 0.4021 | 0.3273 | 0.6473 | 71.62 GB |
| | | EAGLE | **0.6479** | **0.4345** | **0.7039** | **0.5867** | **0.2710** | **0.6840** | **8.75 GB** |
| | Qwen2.5-VL 7B (Bai et al., 2025) | LLaVA-CAM (Zhang et al., 2025b) | 0.5605 | 0.5464 | 0.7235 | 0.4467 | 0.4209 | 0.7010 | 47.17 GB |
| | | iGOS++ (w/ GNC) (Xing et al., 2025) | 0.5603 | 0.5072 | 0.7237 | 0.4400 | 0.3623 | 0.6695 | 96.90 GB |
| | | EAGLE | **0.7006** | **0.4597** | **0.7578** | **0.6337** | **0.2988** | **0.7285** | **17.68 GB** |
| | InternVL3.5 4B (Wang et al., 2025) | LLaVA-CAM (Zhang et al., 2025b) | 0.6116 | 0.6235 | 0.8032 | 0.4948 | 0.5100 | 0.7764 | 81.84 GB |
| | | iGOS++ (w/ GNC) (Xing et al., 2025) | 0.6271 | 0.5726 | 0.7999 | 0.5088 | 0.4337 | 0.7715 | 60.93 GB |
| | | EAGLE | **0.7665** | **0.4650** | **0.8335** | **0.7042** | **0.3042** | **0.8051** | **12.45 GB** |
| MMVP (Tong et al., 2024) (VQA task) | LLaVA-1.5 7B (Liu et al., 2024) | LLaVA-CAM (Zhang et al., 2025b) | 0.7756 | 0.7745 | 0.7980 | 0.6076 | 0.6044 | 0.7275 | 34.38 GB |
| | | iGOS++ (w/ GNC) (Xing et al., 2025) | 0.7717 | 0.7698 | 0.7965 | 0.5825 | 0.5781 | 0.7236 | 92.90 GB |
| | | EAGLE | **0.7960** | **0.7474** | **0.8086** | **0.6867** | **0.5027** | **0.7507** | **15.40 GB** |
| | Qwen2.5-VL 3B (Bai et al., 2025) | LLaVA-CAM (Zhang et al., 2025b) | 0.7742 | 0.7770 | 0.8181 | 0.5925 | 0.6006 | 0.7476 | 19.17 GB |
| | | iGOS++ (w/ GNC) (Xing et al., 2025) | 0.7719 | 0.7613 | 0.8183 | 0.5719 | 0.5356 | 0.7437 | 19.79 GB |
| | | EAGLE | **0.8052** | **0.7338** | **0.8339** | **0.6634** | **0.4935** | **0.7689** | **8.76 GB** |
| | Qwen2.5-VL 7B (Bai et al., 2025) | LLaVA-CAM (Zhang et al., 2025b) | 0.7505 | 0.7486 | 0.8042 | 0.4974 | 0.4847 | 0.7242 | 37.54 GB |
| | | iGOS++ (w/ GNC) (Xing et al., 2025) | 0.7394 | 0.7211 | 0.8036 | 0.4505 | 0.3853 | 0.7185 | 32.76 GB |
| | | EAGLE | **0.7824** | **0.6996** | **0.8119** | **0.5901** | **0.3675** | **0.7362** | **17.40 GB** |
| | InternVL3.5 4B (Wang et al., 2025) | LLaVA-CAM (Zhang et al., 2025b) | 0.7348 | 0.7458 | 0.8325 | 0.4897 | 0.5213 | 0.7575 | 27.20 GB |
| | | iGOS++ (w/ GNC) (Xing et al., 2025) | 0.7277 | 0.7160 | 0.8302 | 0.4743 | 0.4454 | 0.7535 | 62.31 GB |
| | | EAGLE | **0.8012** | **0.6782** | **0.8471** | **0.6379** | **0.4027** | **0.7762** | **12.26 GB** |

decision process. We adopt *Insertion* (Petsiuk et al., 2018), *Deletion* (Petsiuk et al., 2018), and *Average Highest Score* (Chen et al., 2024b), computed as the mean probability over selected tokens. (2) Localization metrics assess whether explanations overlap with ground-truth regions using the *Point Game* (Zhang et al., 2018), under both *box-level* and *mask-level* annotations, where correctness is defined by the maximum attribution point falling inside the bounding box or segmentation mask. (3) Correction-oriented metrics address hallucination evaluation by testing whether attributions reveal regions causing hallucinated outputs. We use *Average Minimal Correction Region (AMCR)*, the average proportion of regions that must be removed to correct hallucinations, and *Correction Success Rate under Budget (CSR@10%)*, the percentage of cases corrected when no more than 10% of regions are removed.

## 4.2 FAITHFULNESS ON SENTENCE-LEVEL EXPLANATIONS

We begin by evaluating our attribution method on two common MLLM tasks, image captioning and visual question answering (VQA), with the goal of identifying which image regions drive the full content generated by the model. We primarily compare our approach against LLaVA-CAM (Zhang et al., 2025b) and IGOS++ (w/ GNC) (Xing et al., 2025). Table 1 reports results on faithfulness metrics, evaluated in two ways: (1) using the sum of logits over all predicted tokens, and (2) using the sum over sensitive tokens, defined as those whose logits change by more than 0.2 when the entire image is masked.

For the image captioning task, our method consistently achieves state-of-the-art performance across all models and metrics. On the LLaVA-1.5 7B model, it surpasses the best results of LLaVA-CAM and IGOS++ (w/ GNC) by 12.7%, 11.9%, and 3.8% in sentence-level insertion, deletion, and average highest score, respectively. At the sensitive-token level, the improvements are even larger, reaching 29.6%, 26.3%, and 3.6%. These stronger gains arise because sensitive tokens are more strongly grounded in visual evidence, making them particularly responsive to well-localized attribution maps. Similar trends are observed on the Qwen2.5-VL 7B model, where our method improves over the best baselines by 25.0%, 9.4%, and 4.7% at the sentence level, and by 41.9%, 17.5%, and 3.9% at the sensitive-token level. On the InternVL3.5 4B model, the corresponding improvements are 22.2%, 18.8%, and 3.8% at the sentence level, and 38.4%, 29.9%, and 3.7% at the sensitive-token level.

For the VQA task, our method also achieves state-of-the-art performance across all models and metrics, though the margins are generally smaller than for captioning. On the LLaVA-1.5 7B model, it improves over the best baselines by 2.6%, 3.0%, and 1.3% at the sentence level, and by 13.0%, 13.0%, and 3.2% at the sensitive-token level. On the Qwen2.5-VL 7B model, the corresponding improvements are 4.3%, 3.0%, and 1.0% at the sentence level, and 18.6%, 1.8%, and 1.7% at the sensitive-token level. On the InternVL3.5 4B model, our method achieves 9.0%, 3.8%, and 1.8% improvements at the sentence level, and 30.3%, 9.6%, and 2.5% at the sensitive-token level. The smaller margins in VQA reflect the fact that much of the generated output relies on reasoning and language priors rather than purely on perceptual evidence.

In addition to higher attribution fidelity, EAGLE demonstrates strong efficiency, requiring only 17.68 GB on Qwen2.5-VL 7B compared to 96.90 GB for IGOS++, making it practical for modern MLLMs. Overall, it provides more faithful and resource-efficient explanations than gradient-based baselines.

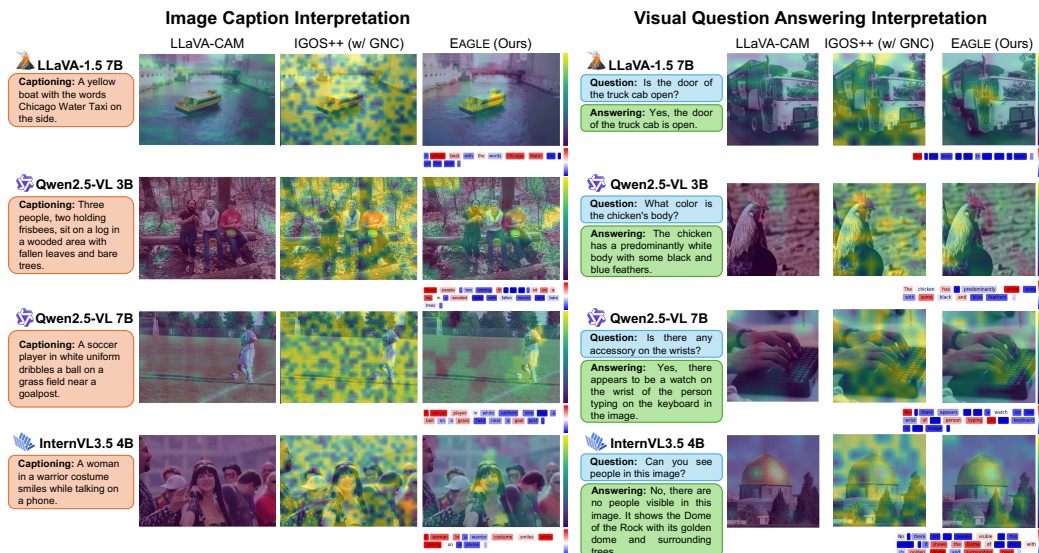

Figure 3: Visualization of explanation results for LLaVA-1.5, Qwen2.5-VL, and InternVL3.5 on the MS COCO and MMVP datasets.

Table 2: Evaluation of word-level faithfulness metrics (Deletion, Insertion AUC, and Average Highest Score) and location metrics (Point Game) on the MS COCO.

| Datasets | MLLMs | Methods | Word-level Faithfulness Metrics | | | Localization Metrics | | GPU Memory (↓) |
|---|---|---|---|---|---|---|---|---|
| | | | Insertion (↑) | Deletion (↓) | Ave. high. score (↑) | Point Game$_{bbox}$ (↑) | Point Game$_{mask}$ (↑) | |
| MS COCO (Lin et al., 2014) (Image caption task) | LLaVA-1.5 7B (Liu et al., 2024) | LLaVA-CAM (Zhang et al., 2025b) | 0.4063 | 0.4035 | 0.6053 | 0.2468 | 0.1168 | 36.73 GB |
| | | iGOS++ (w/ GNC) (Xing et al., 2025) | 0.4093 | 0.3812 | 0.6084 | 0.6623 | 0.5584 | 93.12 GB |
| | | TAM (Li et al., 2025a) | 0.3860 | 0.4162 | 0.5988 | 0.1818 | 0.1428 | 16.60 GB |
| | | EAGLE | 0.6395 | 0.2047 | 0.7213 | 0.8052 | 0.7792 | 16.31 GB |
| | Qwen2.5-VL 3B (Bai et al., 2025) | LLaVA-CAM (Zhang et al., 2025b) | 0.3417 | 0.4575 | 0.7263 | 0.1045 | 0.0621 | 26.01 GB |
| | | iGOS++ (w/ GNC) (Xing et al., 2025) | 0.4141 | 0.2901 | 0.7250 | 0.5822 | 0.4967 | 58.1 GB |
| | | TAM (Li et al., 2025a) | 0.5130 | 0.2797 | 0.7985 | 0.5294 | 0.4379 | 9.56 GB |
| | | EAGLE | 0.7353 | 0.1628 | 0.8641 | 0.8104 | 0.7745 | 9.22 GB |
| | Qwen2.5-VL 7B (Bai et al., 2025) | LLaVA-CAM (Zhang et al., 2025b) | 0.4170 | 0.4771 | 0.8041 | 0.2176 | 0.1428 | 44.26 GB |
| | | iGOS++ (w/ GNC) (Xing et al., 2025) | 0.4816 | 0.3478 | 0.8080 | 0.6734 | 0.5959 | 82.14 GB |
| | | TAM (Li et al., 2025a) | 0.5768 | 0.3167 | 0.8240 | 0.5369 | 0.4060 | 18.75 GB |
| | | EAGLE | 0.8109 | 0.2127 | 0.9194 | 0.7785 | 0.7383 | 18.03 GB |
| | InternVL3.5 4B (Wang et al., 2025) | LLaVA-CAM (Zhang et al., 2025b) | 0.4988 | 0.5040 | 0.8588 | 0.3201 | 0.2212 | 81.84 GB |
| | | iGOS++ (w/ GNC) (Xing et al., 2025) | 0.5192 | 0.3983 | 0.8604 | 0.5775 | 0.5181 | 60.06 GB |
| | | TAM (Li et al., 2025a) | 0.6317 | 0.3517 | 0.8712 | 0.5775 | 0.4653 | 14.23 GB |
| | | EAGLE | 0.8623 | 0.1706 | 0.9585 | 0.8052 | 0.7755 | 7.61 GB |

As shown in Fig. 3, LLaVA-CAM often misses key regions and IGOS++ yields redundant maps, while our method highlights critical regions that align closely with visually grounded tokens, producing concise and human-consistent explanations.

## 4.3 FAITHFULNESS AND LOCALIZATION ON WORD-LEVEL EXPLANATIONS

Next, we evaluate the ability of the proposed attribution method to provide word-level explanations. Specifically, we use samples with object bounding box annotations from the MS COCO dataset to verify whether the objects mentioned in image captions are accurately grounded in the visual input. We also include TAM (Li et al., 2025a) as an additional baseline, since it is particularly effective at explaining object localization.

Table 2 reports the results of faithfulness and localization evaluations, where our method consistently achieves state-of-the-art performance across all models and metrics. For faithfulness, on the LLaVA-1.5 7B model, it surpasses the strongest baseline by 56.2%, 46.3%, and 19.3% in insertion, deletion, and average highest score, respectively. On the Qwen2.5-VL 7B model, the corresponding improvements are 40.6%, 10.4%, and 11.6%, while on the InternVL3.5 4B model, they are 36.5%, 51.5%, and 10.0%. We also observe that TAM performs well only on stronger MLLMs such as Qwen2.5-VL and InternVL3.5, since it relies solely on activation maps rather than capturing strong causal relationships. In contrast, our method is broadly applicable across models and can faithfully explain word-level decisions even for LLaVA-1.5.

For localization, our method achieves the best Pointing Game results under both box- and mask-level settings, confirming that predictions are grounded in specific objects. While TAM performs well on stronger models but poorly on LLaVA-1.5, IGOS++ gains from overly redundant maps. In contrast, our method yields sparse yet focused highlights that more accurately localize the objects mentioned in captions (Fig. 4).

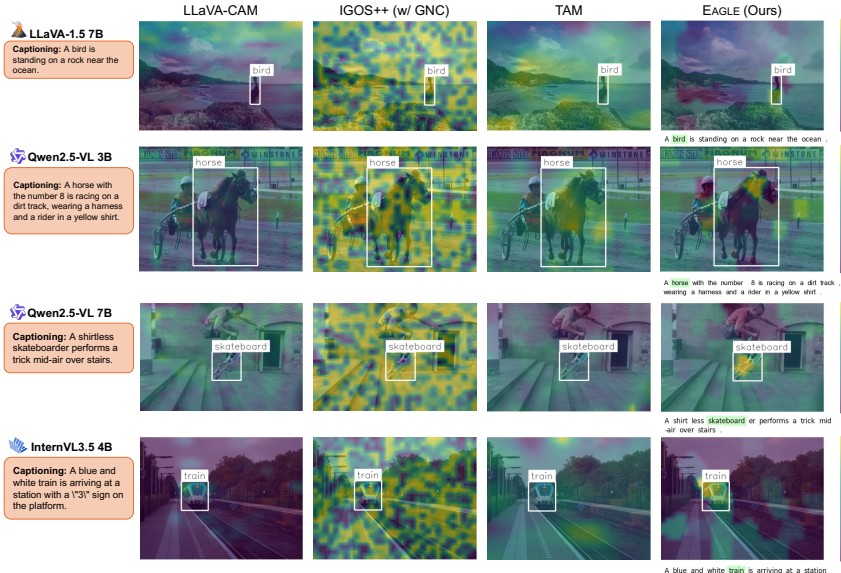

Figure 4: Visualization of word-level explanation results for LLaVA-1.5, Qwen2.5-VL, and InternVL3.5 on the MS COCO datasets.

Table 3: Evaluation of faithfulness metrics (Deletion and Insertion AUC scores) on hallucination interpretation.

| Datasets | MLLMs | Methods | Faithfulness Metrics | | | Correction-oriented Metrics | | GPU Memory (↓) |
|---|---|---|---|---|---|---|---|---|
| | | | Insertion (↑) | Deletion (↓) | Ave. high. score (↑) | AMCR (↓) | CSR@10% (↑) | |
| | LLaVA-1.5 7B (Liu et al., 2024) | LLaVA-CAM (Zhang et al., 2025b) | 0.4095 | 0.4191 | 0.6596 | 0.5613 | 19.70% | 37.07 GB |
| | | iGOS++ (w/ GNC) (Xing et al., 2025) | 0.4232 | 0.4182 | 0.6794 | 0.4770 | 37.50% | 93.88 GB |
| | | TAM (Li et al., 2025a) | 0.4168 | 0.4166 | 0.6705 | 0.5826 | 18.75% | 16.59 GB |
| | | EAGLE | **0.6999** | **0.2652** | **0.7877** | **0.0844** | **77.50%** | **16.04 GB** |
| RePOPE (Neuhaus & Hein, 2025) (Object Hallucination Benchmark) | Qwen2.5-VL 3B (Bai et al., 2025) | LLaVA-CAM (Zhang et al., 2025b) | 0.3994 | 0.3783 | 0.6992 | 0.4555 | 42.21% | 27.10 GB |
| | | iGOS++ (w/ GNC) (Xing et al., 2025) | 0.4056 | 0.4471 | 0.7235 | 0.4461 | 37.57% | 35.16 GB |
| | | TAM (Li et al., 2025a) | 0.3905 | 0.4090 | 0.6900 | 0.4747 | 29.75% | 9.66 GB |
| | | EAGLE | **0.7568** | **0.1610** | **0.8717** | **0.0849** | **80.41%** | **9.20 GB** |
| | Qwen2.5-VL 7B (Bai et al., 2025) | LLaVA-CAM (Zhang et al., 2025b) | 0.2444 | 0.2901 | 0.5898 | 0.6620 | 35.37% | 45.05 GB |
| | | iGOS++ (w/ GNC) (Xing et al., 2025) | 0.3017 | 0.3330 | 0.7125 | 0.5357 | 32.41% | 70.86 GB |
| | | TAM (Li et al., 2025a) | 0.2717 | 0.3177 | 0.6792 | 0.5844 | 22.45% | 18.57 GB |
| | | EAGLE | **0.7987** | **0.0331** | **0.9381** | **0.1442** | **73.94%** | **18.26 GB** |
| | InternVL3.5 4B (Wang et al., 2025) | LLaVA-CAM (Zhang et al., 2025b) | 0.4079 | 0.3733 | 0.9296 | 0.4078 | 36.43% | 87.93 GB |
| | | iGOS++ (w/ GNC) (Xing et al., 2025) | 0.3651 | 0.4556 | 0.9393 | 0.4299 | 38.76% | 66.26 GB |
| | | TAM (Li et al., 2025a) | 0.3794 | 0.4221 | 0.9115 | 0.4801 | 28.57% | 14.04 GB |
| | | EAGLE | **0.9114** | **0.0440** | **0.9941** | **0.0676** | **80.00%** | **12.31 GB** |

## 4.4 INTERPRETING OBJECT HALLUCINATION

We next apply our interpretable algorithm to analyze why MLLMs produce hallucinations. Experiments are conducted on the object hallucination benchmark RePOPE (Neuhaus & Hein, 2025). We focus on samples where the MLLM makes prediction errors, including cases where the model incorrectly answers "no" instead of "yes," and vice versa. Assuming that hallucinations have already been identified, our objective is to identify which image regions trigger the hallucination and to assess whether blocking these regions can mitigate it. In practice, we attribute the first token of the answer, restricted to the vocabulary IDs 'Yes' and 'No'. For example, if the model incorrectly outputs 'Yes', the attribution is computed with respect to 'No', thereby providing a counterfactual perspective on which regions would support the correct response.

Table 3 reports the results of attributing hallucinations to specific input regions. On the LLaVA-1.5 7B model, our method improves over the strongest baseline by 65.4%, 36.3%, and 15.9% in insertion, deletion, and average highest score, respectively. On the Qwen2.5-VL 7B model, the gains are even larger, reaching 164.7%, 88.6%, and 31.7%, while on the InternVL3.5 4B model, the improvements are 123.4%, 88.2%, and 5.8%. These substantial margins highlight the strength of our approach in faithfully uncovering the input regions responsible for hallucinated predictions and in explaining the underlying causes of incorrect decisions, revealing not only where the model looked, but also why it went wrong.

Next, we examine whether hallucinations can be eliminated by progressively removing the responsible regions. Instead of relying on logits, we evaluate direct model outputs (Yes or No with the corresponding rationale) using correction-oriented metrics. On the LLaVA-1.5 7B model, our method surpasses the strongest baseline by 82.3% and 106.6% in Average Minimal Correction Region (AMCR) and Correction Success Rate under Budget (CSR@10%), respectively. On the Qwen2.5-VL 7B model, the improvements are 73.1% and 109.0%, and on the InternVL3.5 4B model they are 83.4% and 106.4%. These results show that removing only a small portion of the input is sufficient to eliminate hallucinations, demonstrating the effectiveness of our attribution approach.

Table 4: Ablation of objective function components on Qwen2.5-VL 7B for MS COCO captioning.

| Insight (Eq. 2) | Necessity (Eq. 3) | Faithfulness Metrics | | |
|---|---|---|---|---|
| | | Ins. ($\uparrow$) | Del. ($\downarrow$) | Avg. High ($\uparrow$) |
| ✗ | ✓ | 0.6176 | 0.4613 | 0.7282 |
| ✓ | ✗ | 0.6981 | 0.5253 | 0.7566 |
| ✓ | ✓ | **0.7006** | **0.4597** | **0.7578** |

Table 5: Ablation of subregion number on Qwen2.5-VL 7B for MS COCO captioning.

| Number | Faithfulness Metrics | | |
|---|---|---|---|
| | Ins. ($\uparrow$) | Del. ($\downarrow$) | Avg. High ($\uparrow$) |
| 36 | 0.6869 | 0.4587 | 0.7452 |
| 50 | 0.6901 | **0.4514** | 0.7482 |
| 64 | **0.7006** | 0.4597 | **0.7578** |

Figure 5: Hallucination attribution on RePOPE. Our method produces sparse, focused maps that more accurately reveal regions responsible for hallucinated outputs, compared with IGOS++ and TAM.

Fig. 5 visualizes the results, including the Hallucination Map, where highlighted purple regions indicate areas prone to hallucinations identified by our method. Hallucination Mitigation denotes the minimal region that must be removed to eliminate hallucinations. The curve illustrates changes in the logit of the ground-truth token as hallucination-prone regions are progressively deleted, with the red line marking the deletion point determined by Hallucination Mitigation. Our method rapidly localizes regions that cause hallucinations, while TAM and IGOS++ produce diffuse maps. On LLaVA-1.5, it attributes the false detection of a snowboard to a surfboard, highlighting confusion between similar objects. InternVL3.5 fails to recognize a spoon that is partially occluded by a fork. By precisely attributing and removing the fork head, our method enables the model to correctly identify the spoon, revealing its limited ability to disambiguate overlapping objects.

### 4.5 ABLATION STUDY

We conduct ablations on the MS COCO captioning task with Qwen2.5-VL 7B to evaluate both the objective function design and the impact of subregion partitioning. As shown in Table 4, only the joint use of the Insight and Necessity Scores consistently improves all faithfulness metrics, demonstrating their complementary effects. Table 5 further shows that finer image partitions generally enhance faithfulness, though at the expense of increased attribution time, suggesting the importance of developing more scalable attribution strategies in future work.

## 5 CONCLUSION AND LIMITATION

In this paper, we present EAGLE, a black-box attribution framework for autoregressive MLLMs. By unifying sufficiency and indispensability in a submodular-inspired objective, EAGLE faithfully explains token generation, revealing both *where* models attend and *what* they rely on. Experiments across diverse models and datasets show clear gains in faithfulness, localization, and hallucination diagnosis. Moreover, by identifying and removing minimal interfering regions, EAGLE also mitigates hallucinations, serving as both an interpretability and correction-oriented tool.

**Limitations.** Despite its effectiveness, our work has two main limitations. First, the iterative subset selection and greedy search limit scalability compared to lightweight visualization methods. Second, the framework focuses on hallucination explanation and partial mitigation, leaving proactive prevention unexplored. Future work will explore faster search strategies and explanation-guided debiasing for training MLLMs.

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

## A  LLM USAGE

During the preparation of this manuscript, large language models (LLMs) were employed in a limited and auxiliary capacity. Specifically, their usage was restricted to the following three aspects: (1) checking grammar and expression at the sentence level, thereby providing local linguistic refinement; (2) performing global polishing after the draft was completed, ensuring that the overall exposition conforms to idiomatic English usage.

At no stage were LLMs used for generating research ideas, developing arguments, or modifying the substantive content of this work. Their sole role was to assist in enhancing the clarity and effectiveness of communication.

## B  EAGLE ALGORITHM

The detailed calculation process of the proposed EAGLE algorithm is outlined below.

---

**Algorithm 1:** EAGLE: Explaining Autoregressive Generation by Language priors or Evidence in multimodal large language models (MLLMs)

---

**Input:** Image $\mathbf{I} \in \mathbb{R}^{h \times w \times 3}$, partitioning algorithm $\texttt{Div}(\cdot)$, prompt $\texttt{Prompt}$, generated sequence $\mathbf{y}$, target token positions $T$, vocabulary indices $\mathcal{V}$.

**Output:** Ordered subset $\pi$, saliency map $\mathcal{A} \in \mathbb{R}^{h \times w}$, influence scores $I_t$.

1  $V \leftarrow \texttt{Div}(\mathbf{I})$;
2  $\pi \leftarrow \varnothing$ ;                              /* Initialize ordered subset */
3  $\mathcal{A}_1 \leftarrow 0$;
4  **for** $i = 1$ **to** $|V|$ **do**
5   $\quad S_d \leftarrow V \setminus S$;
6   $\quad \alpha \leftarrow \arg\max_{\alpha \in S_d} \mathcal{F}(\pi \cup \{\alpha\})$;
7   $\quad \pi \leftarrow \pi \parallel \{\alpha\}$;
8   $\quad$ **if** $i > 1$ **then**
9    $\quad\quad \mathcal{A}_i \leftarrow \mathcal{A}_{i-1} - \left| \mathcal{F}(\pi_{:i}) - \mathcal{F}(\pi_{:i-1}) \right|$ ;           /* Saliency update */
10 **end**
11 **for** $i = 1$ **to** $|T|$ **do**
12  $\quad s_{\max} \leftarrow \max_{1 \le j \le |\pi|} p(y_{t_i} = v_i \mid \pi_{:j}, \texttt{Prompt}, \mathbf{y}_{<t_i})$;
13  $\quad I_{t_i} \leftarrow \sum_{r=1}^{|\pi|} \left( s_{\max} - p(y_{t_i} = v_i \mid \pi_{:r}, \texttt{Prompt}, \mathbf{y}_{<t_i}) \right)$ ;  /* Language prior vs.
   perception evidence */
14 **end**
15 **return** $\pi$, $\text{norm}(\mathcal{A})$, $\text{norm}(I_t)$

---

## C  ADDITIONAL EXPERIMENTAL DETAILS

For the image captioning task on MS COCO, the prompt used for all MLLMs is:

```
Describe the image in one factual English sentence of no more than
20 words.  Do not include information that is not clearly visible.
```

For the hallucination detection task on RePOPE, the prompt used is:

```
You are asked a visual question answering task.
First, answer strictly with "Yes" or "No".
Then, provide a short explanation if necessary.

Question: {question}
Answer:
```

# D ADDITIONAL QUALITATIVE RESULTS

In this appendix, we provide extended qualitative visualizations that complement the main findings in Fig. 3, Fig. 4, and Fig. 5. These supplementary results aim to offer a finer-grained perspective on how competing attribution methods and our proposed approach behave across diverse settings. Specifically, we present: (i) sentence-level explanations on both MS COCO and MMVP, (ii) object-level explanations on MS COCO, and (iii) hallucination attribution visualizations on additional samples. Collectively, these results provide deeper insights into the consistency, precision, and interpretability of our method.

## D.1 SENTENCE-LEVEL EXPLANATIONS ON MS COCO AND MMVP

As shown in Fig. 6 and Fig. 7, our method produces faithful explanations for **LLaVA-1.5** by tightly aligning highlighted regions with relevant caption tokens (e.g., "smiling," "hat," "motor") or VQA queries (e.g., "Is the shark's belly visible?"). In contrast, LLaVA-CAM often distributes attention diffusely across the scene, while IGOS++ over-activates irrelevant background regions.

For **Qwen2.5-VL**, Fig. 8 and Fig. 9 show that our method generates concise and semantically meaningful attribution maps. For example, in captions mentioning multiple objects, our approach selectively highlights the relevant ones while avoiding redundancy. In VQA tasks, it accurately isolates queried entities such as a remote button, whereas baselines either miss the target or introduce noise.

Similarly, for **InternVL3.5** (Fig. 10, Fig. 11), our method highlights precise object-centric regions corresponding to key caption tokens (e.g., "sandwich," "frisbee") and VQA queries (e.g., "Does the snowman have arms made of branches?"). Baseline methods either scatter attention broadly or fail to capture the queried object, reducing interpretability. These results collectively demonstrate that our approach consistently improves faithfulness and transparency across different models and datasets.

## D.2 OBJECT-LEVEL EXPLANATIONS ON MS COCO

Beyond sentence-level results, we further evaluate our method at the object level with ground-truth bounding boxes. Fig. 12, Fig. 13, and Fig. 14 illustrate that our method produces sparse yet highly accurate localization of queried objects such as "boat," "keyboard," or "truck." By contrast, IGOS++ frequently covers overly broad regions, while LLaVA-CAM and TAM often fail to precisely localize objects. These comparisons highlight the advantage of our method in generating interpretable, object-centric attributions.

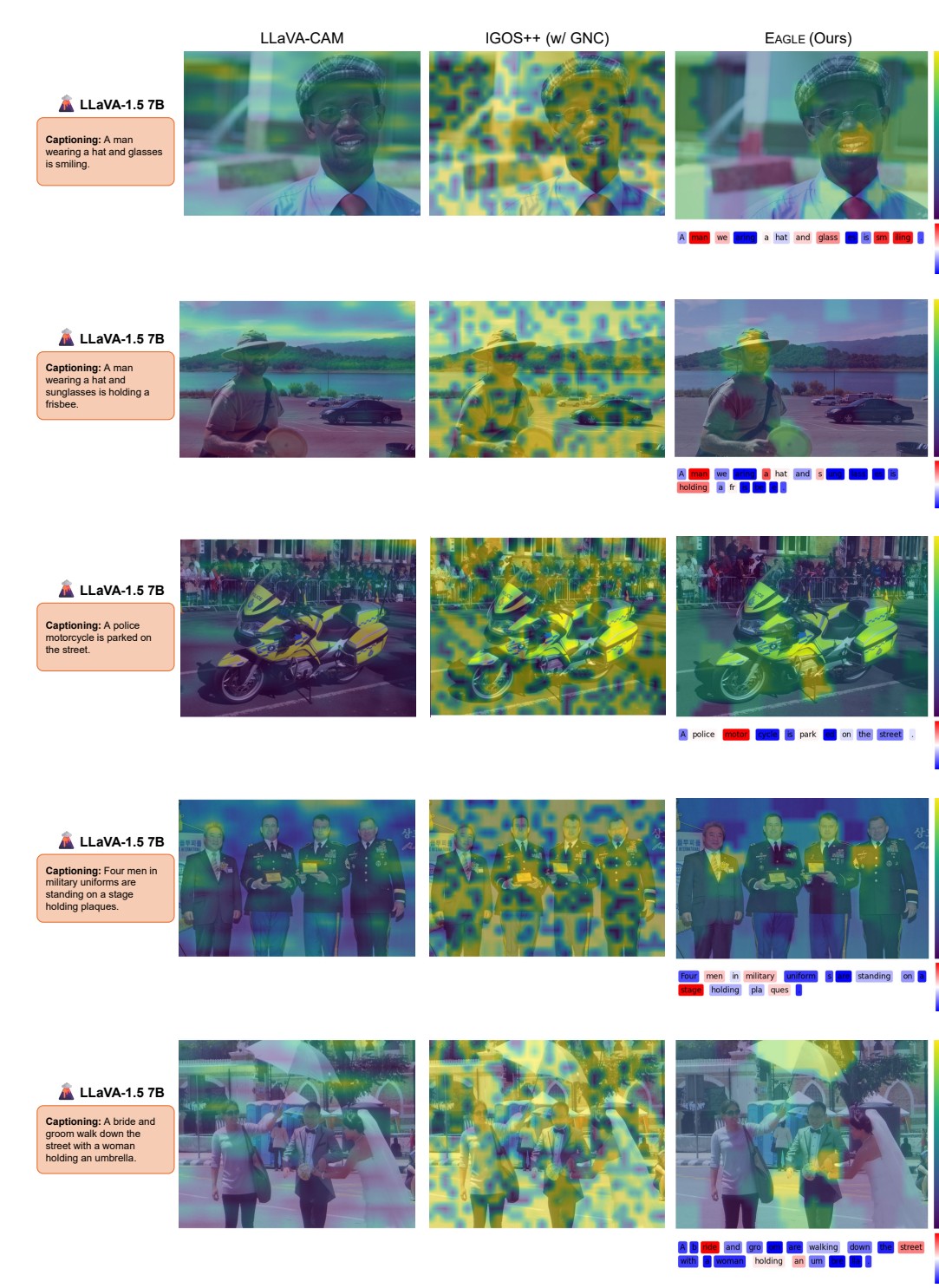

Figure 6: Sentence-level explanation results for **LLaVA-1.5** on the MS COCO dataset. Our method consistently identifies semantically critical regions that align with highlighted tokens in the caption, while baseline methods either fail to capture relevant areas (LLaVA-CAM) or over-highlight irrelevant background regions (IGOS++).

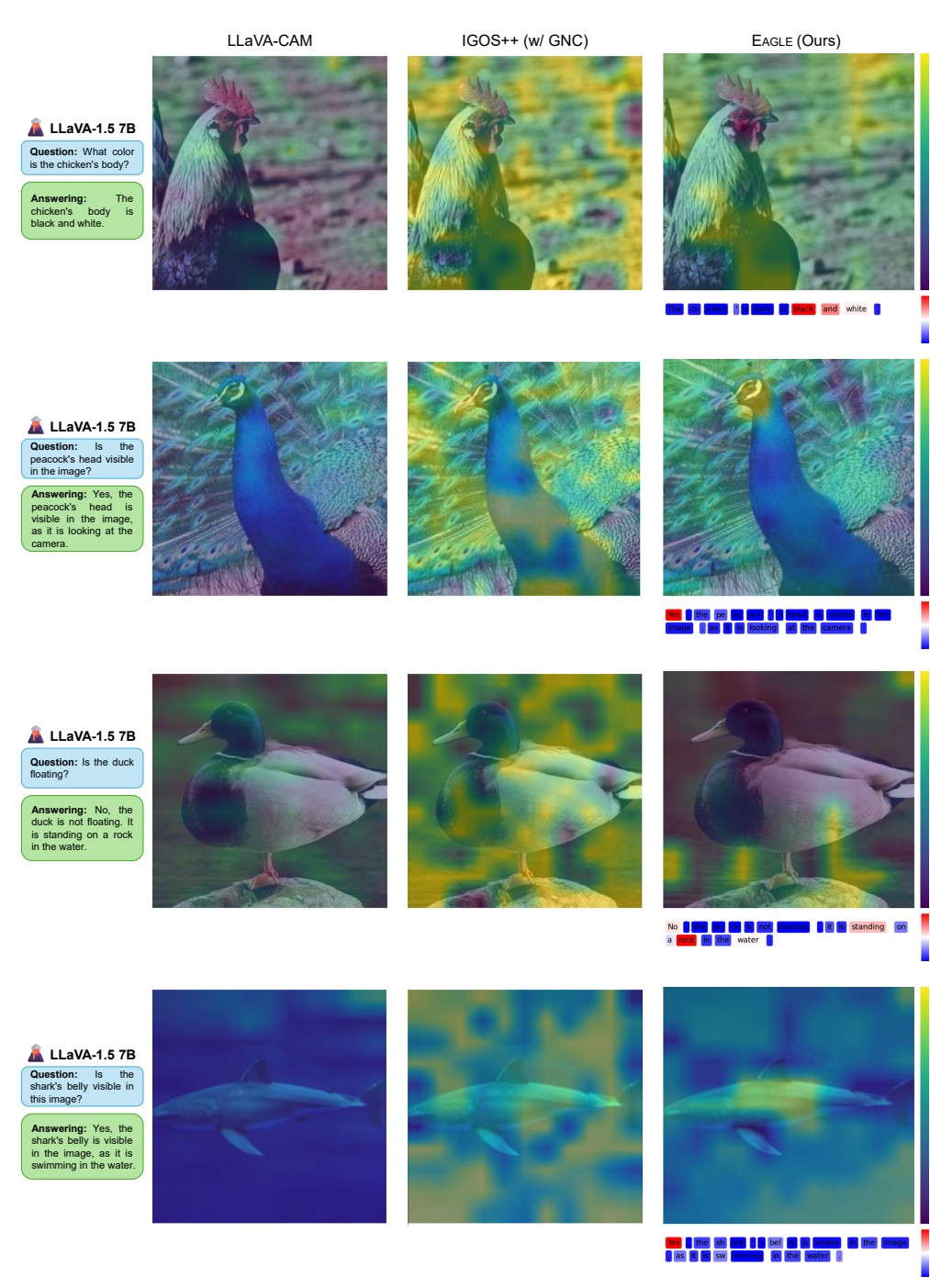

Figure 7: Sentence-level explanation results for **LLaVA-1.5** on the MMVP dataset. Compared to the baselines, our method highlights regions that are directly related to the VQA queries, resulting in explanations that are more interpretable and trustworthy.

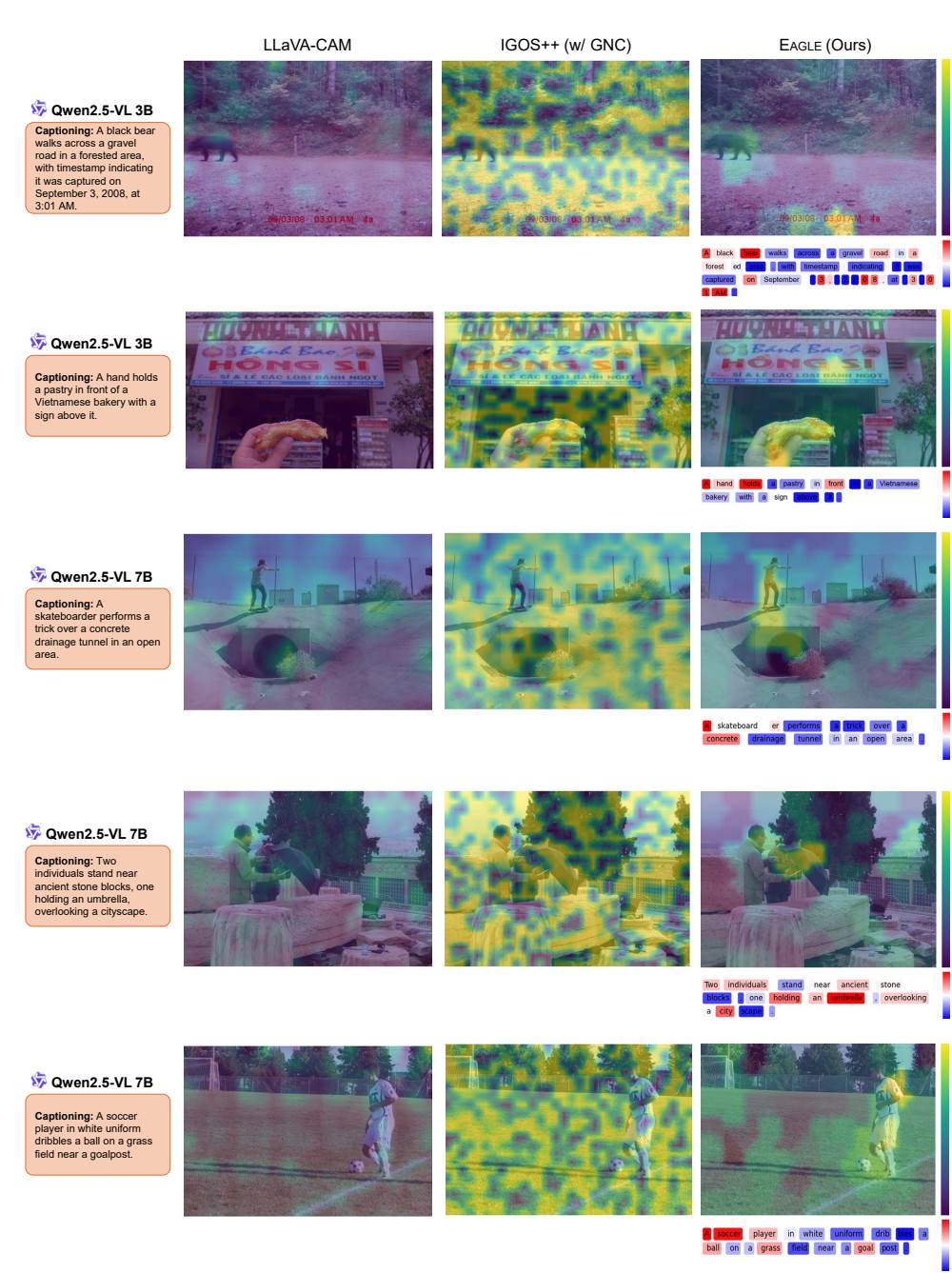

Figure 8: Sentence-level explanation results for **Qwen2.5-VL** on the MS COCO dataset. Our method highlights critical objects with strong correspondence to the generated captions, reducing redundancy in comparison to IGOS++.

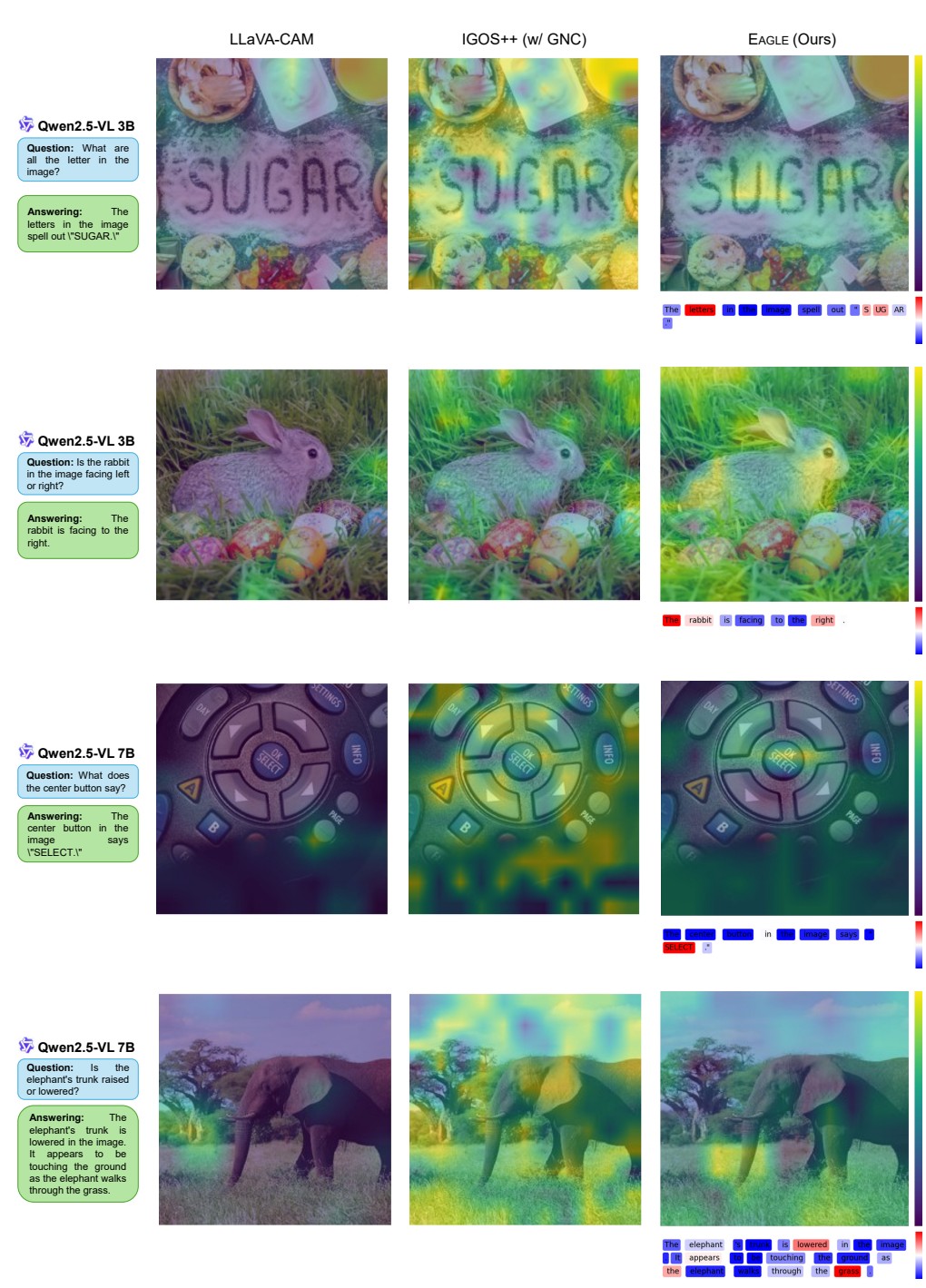

Figure 9: Sentence-level explanation results for **Qwen2.5-VL** on the MMVP dataset. Our method improves alignment between highlighted visual regions and VQA-relevant words, enhancing interpretability.

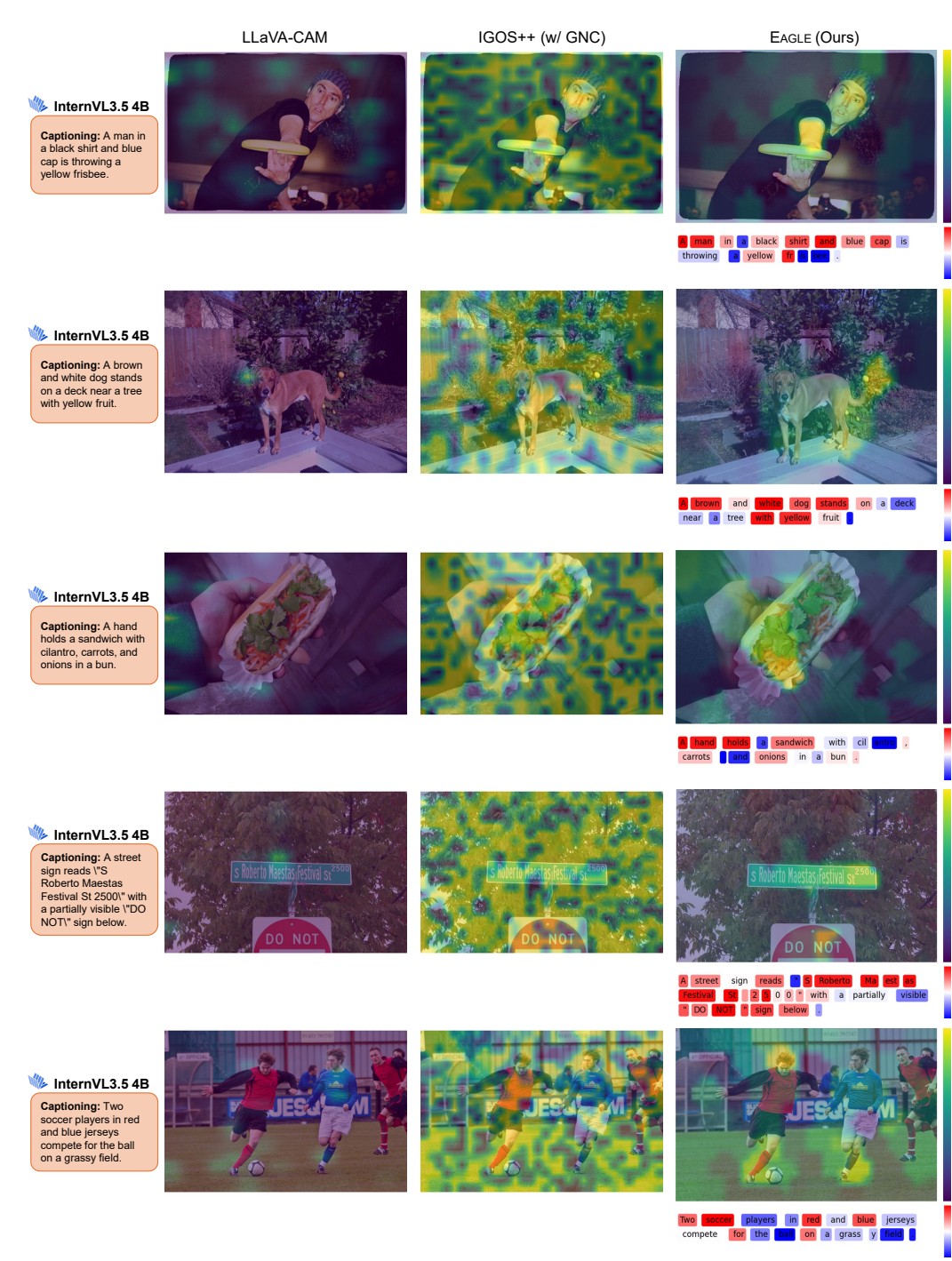

Figure 10: Sentence-level explanation results for **InternVL3.5** on the MS COCO dataset. Our method captures object-centric regions more consistently than baseline methods.

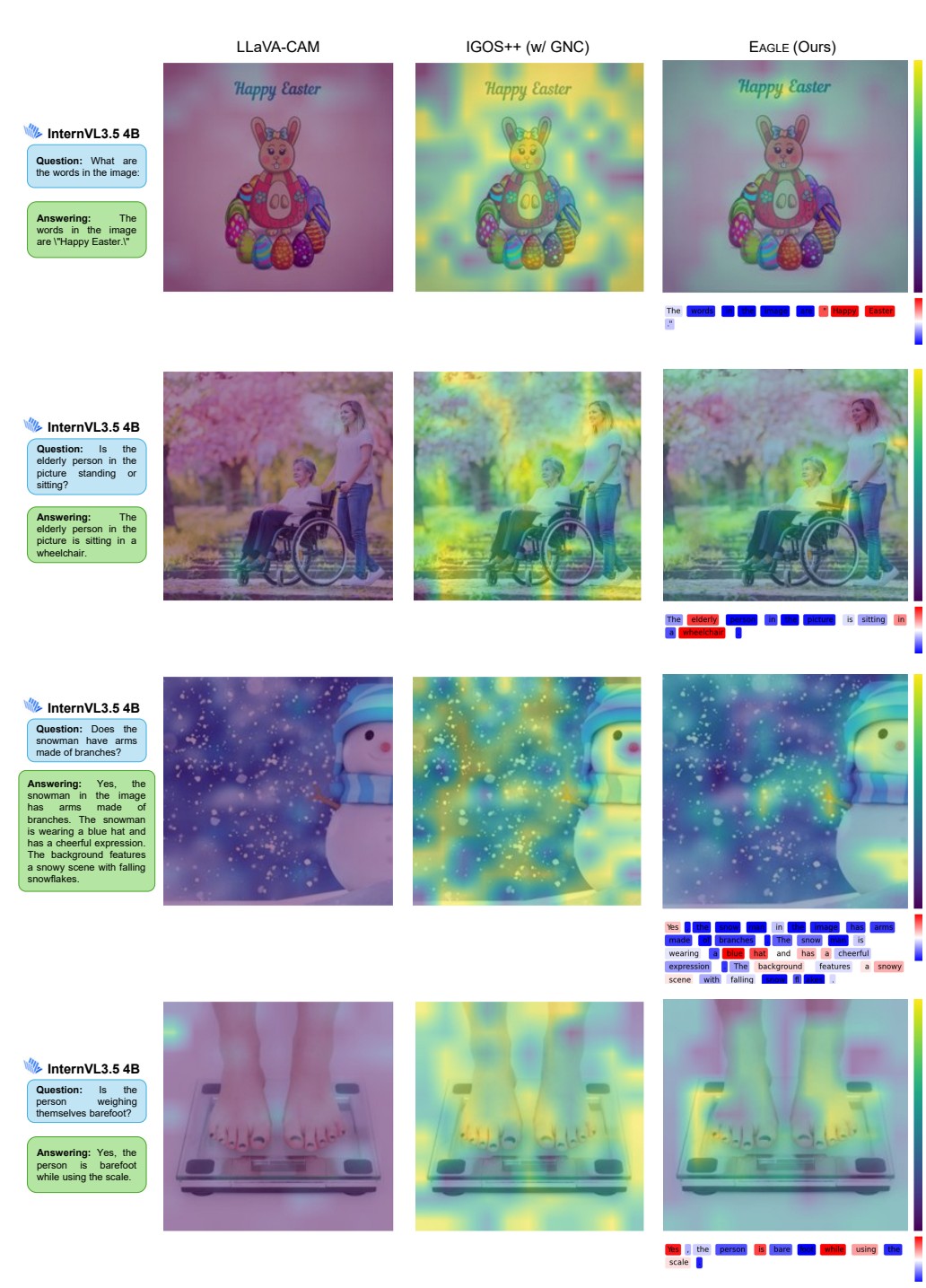

Figure 11: Sentence-level explanation results for **InternVL3.5** on the MMVP dataset. Our approach ensures strong consistency between highlighted evidence and the VQA queries.

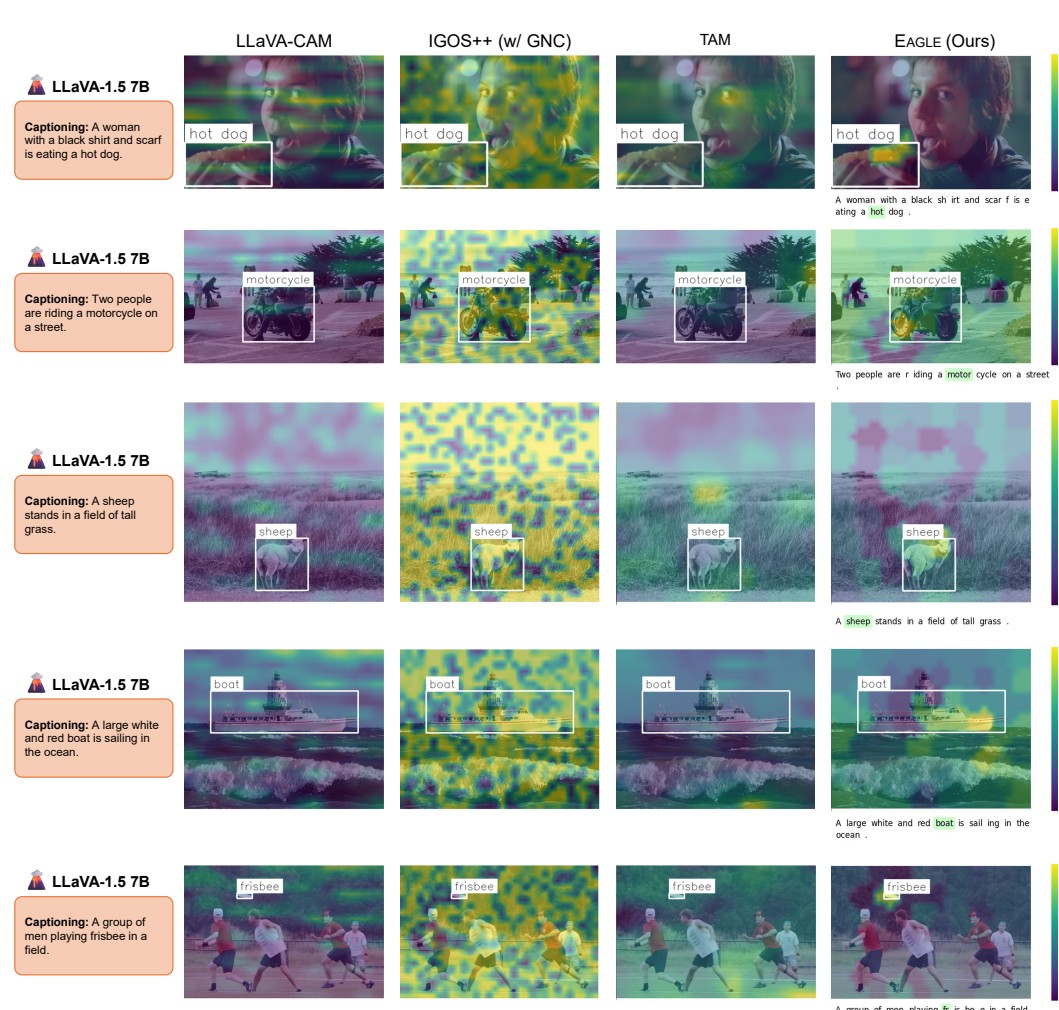

Figure 12: Object-level explanation results for **LLaVA-1.5** on the MS COCO dataset. Bounding box overlays show that our method provides sparse yet highly accurate localization.

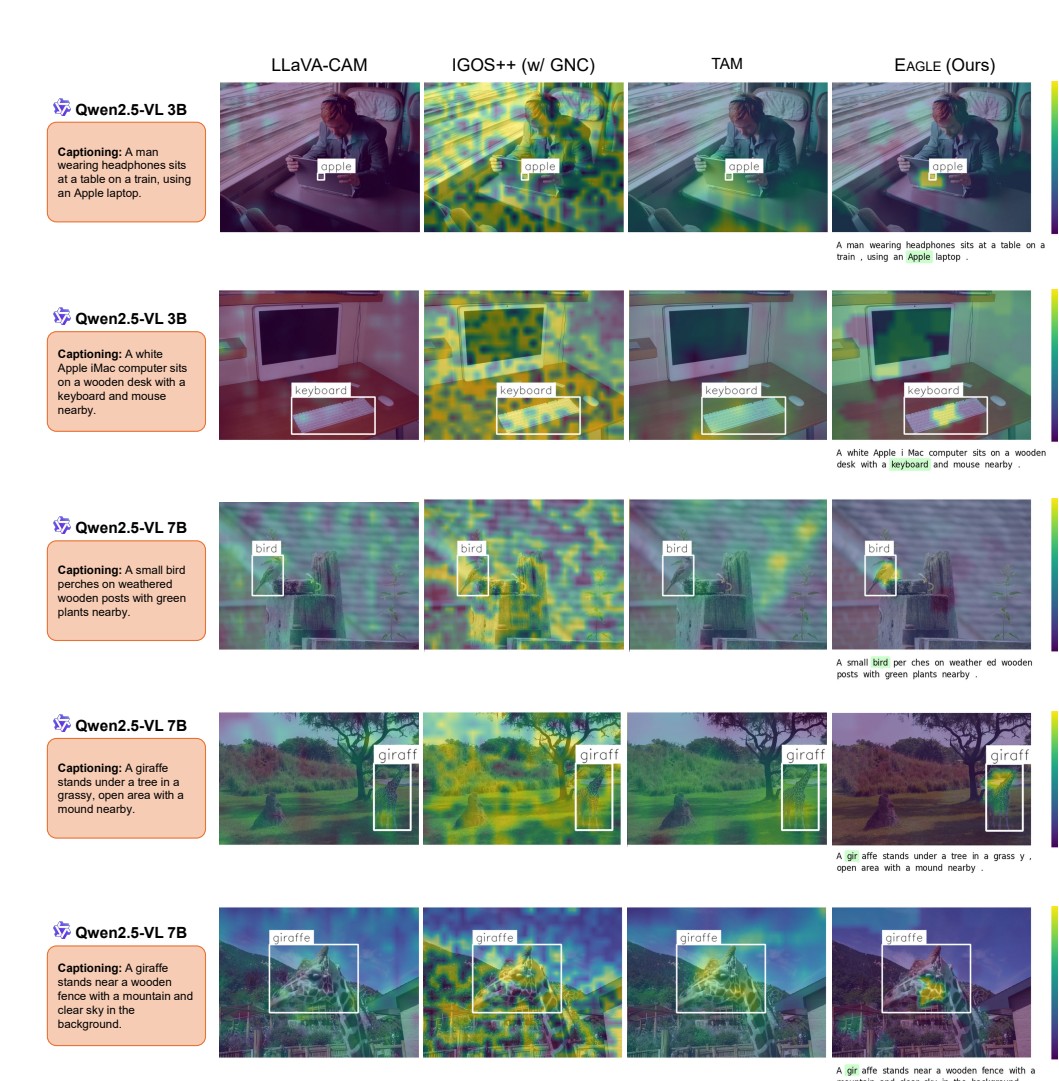

Figure 13: Object-level explanation results for **Qwen2.5-VL** on the MS COCO dataset. Our method produces localized attribution maps with high correspondence to ground-truth bounding boxes.

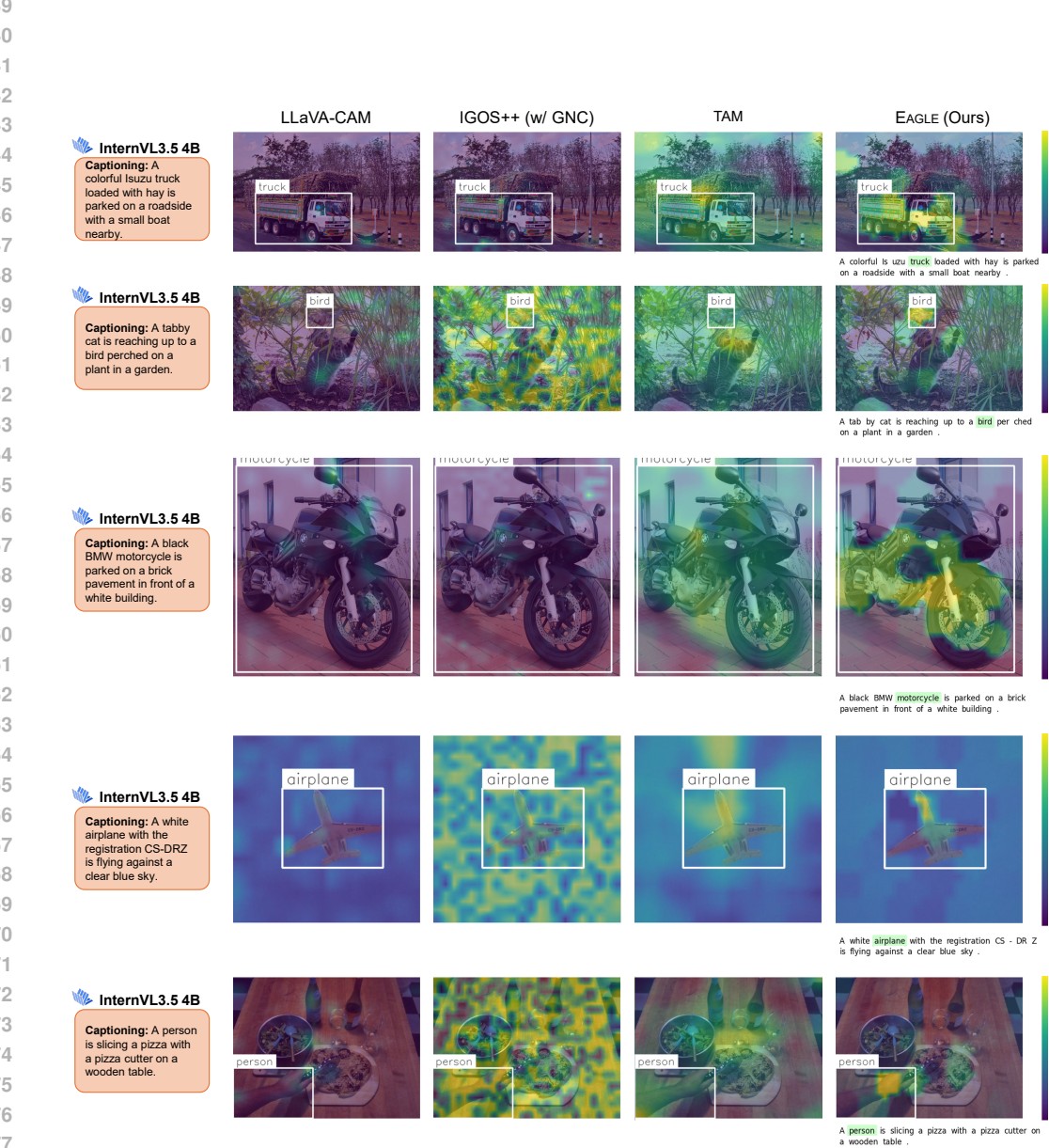

Figure 14: Object-level explanation results for **InternVL3.5** on the MS COCO dataset. Our method captures object-centric highlights with strong correspondence to caption tokens and bounding boxes.

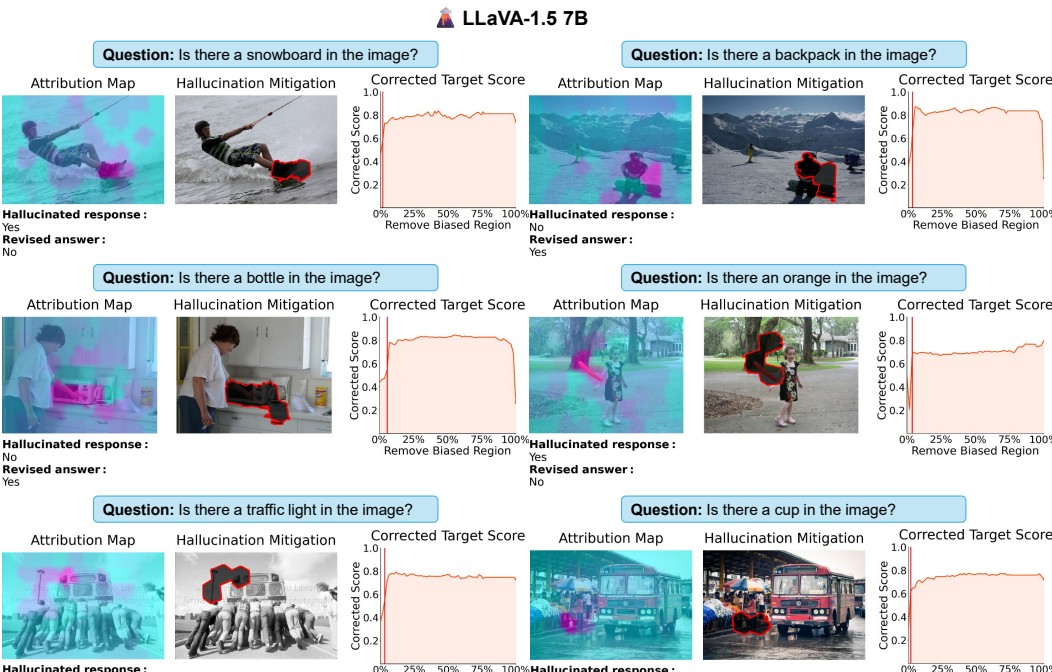

Figure 15: Hallucination attribution for **LLaVA-1.5** on the MS COCO dataset. Our method highlights the minimal hallucination-inducing regions across different queries, such as "snowboard," "traffic light," and "cup."

## D.3 ADDITIONAL HALLUCINATION ATTRIBUTION VISUALIZATIONS

We also provide supplementary hallucination attribution results on MS COCO (Fig. 15, Fig. 16, Fig. 17). Unlike the main paper, these figures focus exclusively on our method to illustrate how it identifies hallucination-prone regions across diverse queries.

For **LLaVA-1.5** (Fig. 15), hallucinations typically arise from visually similar structures. For example, queries about a "snowboard" lead to confusions with surfboard-like regions, while small background cues induce false detections for "traffic light" or "cup." Our attribution maps isolate these exact regions, providing interpretable evidence of failure modes.

For **Qwen2.5-VL** (Fig. 16), hallucinations are often caused by small or occluded objects. For instance, reflective regions resembling a phone screen mislead the model when asked about "cell phones," while circular patterns in the background induce false positives for "bicycle." Our approach sharply localizes these misleading cues, enhancing transparency.

Finally, for **InternVL3.5** (Fig. 17), hallucinations are triggered by overlapping or occluded objects. For example, confusion between a fork and a spoon is precisely localized, as are reflective regions falsely identified as "TVs" or cluttered areas misinterpreted as "dining tables." These examples underscore the effectiveness of our method in diagnosing hallucination sources in a fine-grained and transparent manner.

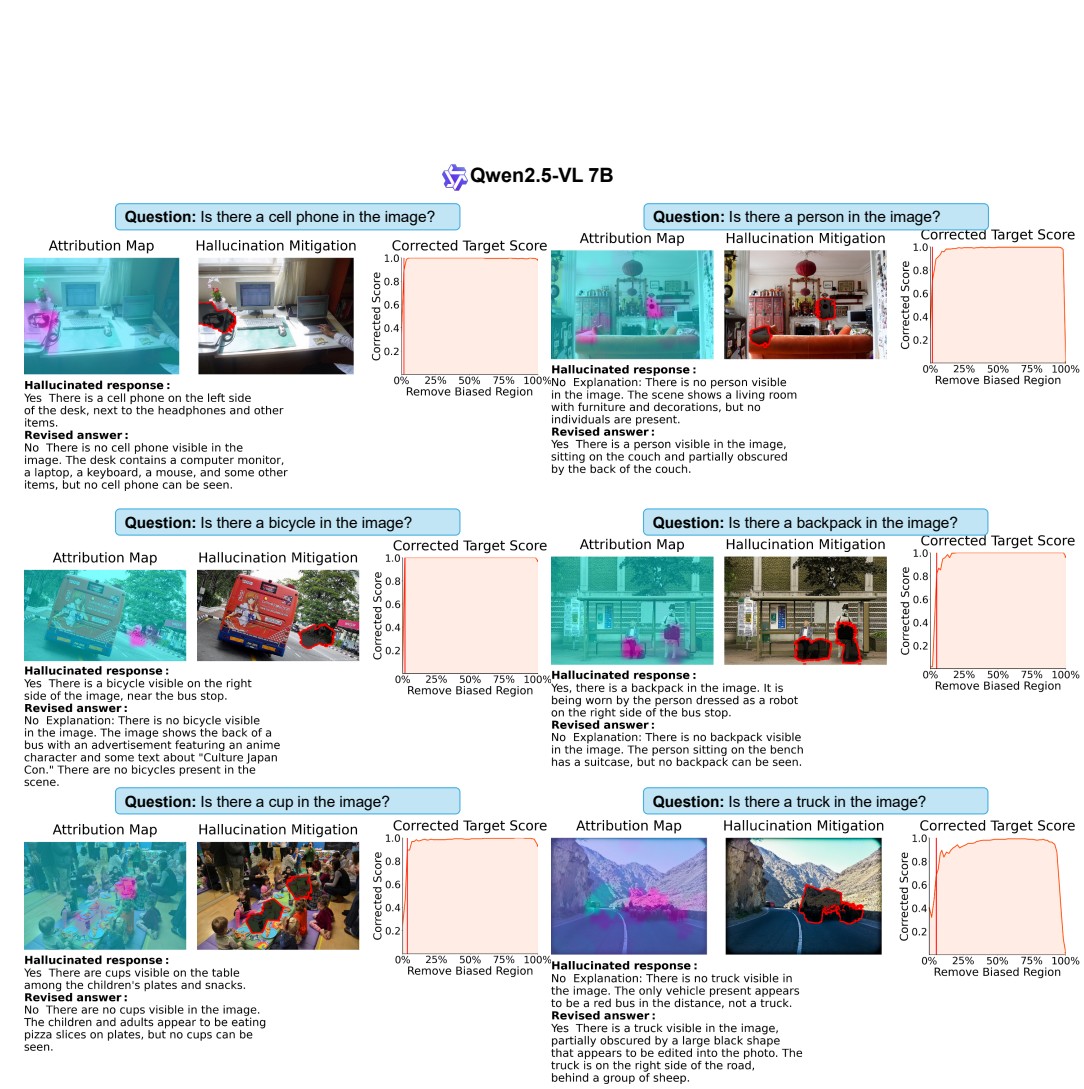

Figure 16: Hallucination attribution for **Qwen2.5-VL** on the MS COCO dataset. Our method isolates misleading cues leading to hallucinations in queries such as "cell phone," "bicycle," and "truck."

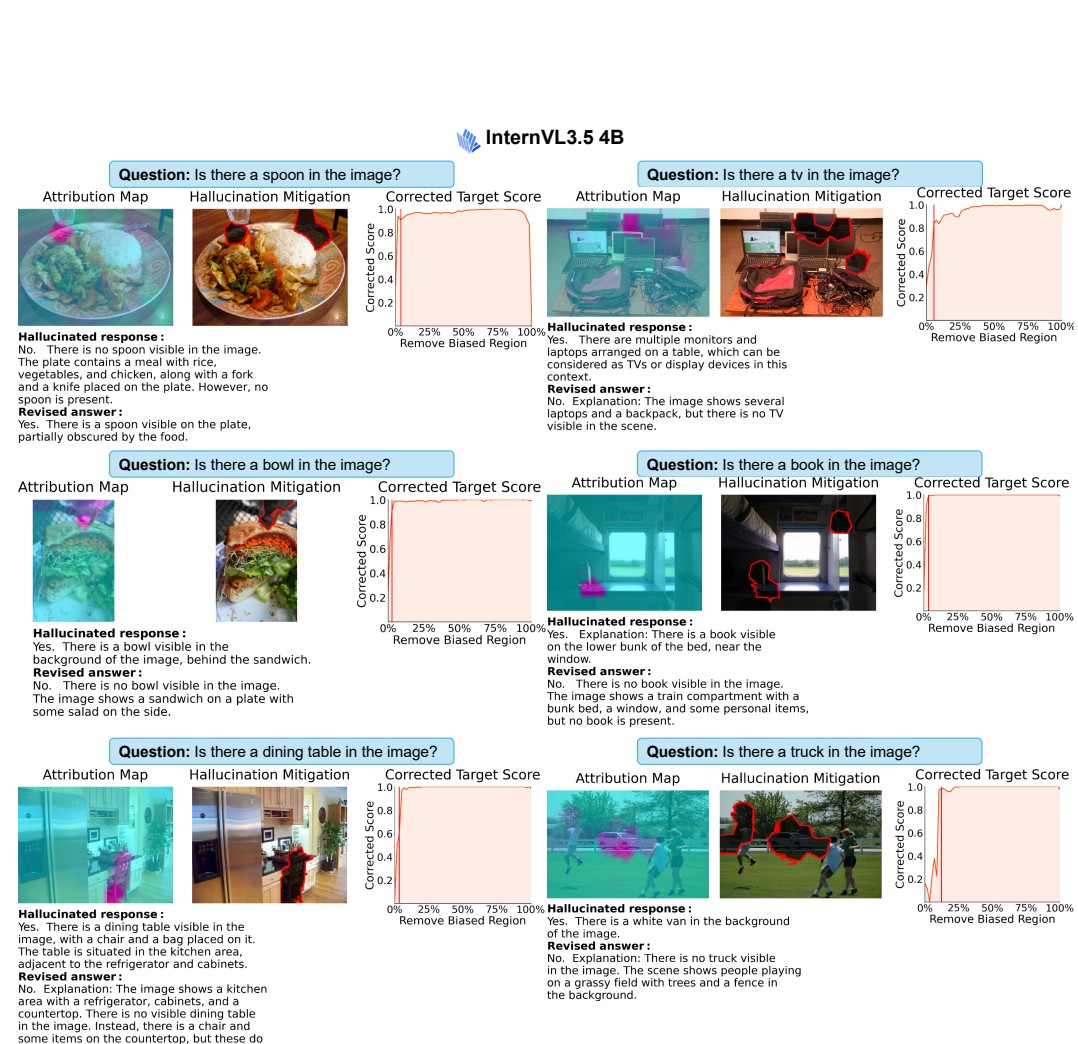

Figure 17: Hallucination attribution for **InternVL3.5** on the MS COCO dataset. Our method identifies hallucination-prone regions for queries such as "spoon," "tv," and "dining table," especially in cases of overlapping or occluded objects.

