# OpenReview forum: "Where MLLMs Attend and What They Rely On: Explaining Autoregressive Token Generation"
_ICLR.cc/2026/Conference — ICLR 2026 Conference Withdrawn Submission_

### Official Review · Reviewer_DF4W · 2025-10-27

**Soundness:** 3
**Presentation:** 2
**Contribution:** 2
**Rating:** 6
**Confidence:** 2

**Summary:**

This paper presents EAGLE, a black-box attribution framework designed to interpret the token generation process of multimodal large language models (MLLMs). The method decomposes an image into superpixel subregions and employs a greedy search over ordered subsets to identify regions that are both sufficient and necessary for generating target tokens. The unified objective function combines an insight score (for sufficiency) and a necessity score (for indispensability). Additionally, the framework introduces a modality analysis that quantifies whether each token is primarily driven by visual evidence or language priors. Extensive experiments across multiple MLLMs demonstrate consistent improvements over gradient- and activation-based baselines on faithfulness, localization, and hallucination benchmarks.

**Strengths:**

The method is intuitive and straightforward, relying on interpretable probability changes on subregions rather than gradients or activations. Experiments are comprehensive and well-designed, covering multiple models and benchmarks. Results show consistent improvements.

**Weaknesses:**

However, I am not familiar with the XAI literature, so I am not fully confident about the method’s novelty relative to prior work. The proposed approach also depends on the quality of the image partitioning algorithm, which could affect stability and interpretability. Moreover, the greedy search requires a full forward pass through the MLLM at each iteration, potentially leading to very high computational cost compared to activation- or gradient-based baselines.

**Questions:**

See Weaknesses.

**Details Of Ethics Concerns:**

N.A.

---

> ### Author Response · Authors · 2025-11-14
> **Official Response to Reviewer DF4W**
>
> We would like to sincerely thank the reviewer `DF4W` for recognizing the value of our work, particularly in how EAGLE provides a novel framework for explaining multimodal large language models (MLLMs). We greatly appreciate the reviewer’s acknowledgment of our method’s intuitiveness, comprehensiveness, and the consistent improvements shown in our experiments across multiple models and benchmarks.
>
> ---
>
> **W1 (Novelty compare to prior work):** We appreciate the reviewer’s concern. Attribution work for MLLMs is limited, with existing methods like LLaVA-CAM, IGOS++, and TAM. The key challenge with MLLMs is their autoregressive generation nature, meaning traditional attribution methods cannot be directly applied. Our method goes beyond visual attribution by also providing modality-level explanations of the output, which is a key innovation not addressed by previous methods. By analyzing the interaction between visual evidence and language priors, we offer insights into how both modalities contribute to token generation, a capability that traditional attribution approaches do not provide. This makes our framework more comprehensive and unique in the field of MLLM interpretability.
>
> ---
>
>
>
> **W2 (Impact on division quality):** We agree that the quality of image partitioning can impact attribution results. However, our method is not dependent on a specific segmentation algorithm and can work with any suitable region partitioning method, such as SLICO or square grids. Previous work has shown that superpixel-based segmentation methods tend to yield higher faithfulness in subset selection, which is why we chose to use it in our experiments. We will include additional comparisons of different partitioning methods for MLLMs in future versions of the paper to explore their impact on attribution results.
>
>
>
> ---
>
> **W3 (High computational cost compared to activation- or gradient-based baselines):** Yes, it is true that the search-based mechanism in our method is slower compared to activation- or gradient-based approaches. However, an increasing body of literature has demonstrated that gradient- and activation-based interpretability methods struggle to explain more complex multimodal large models, as they are primarily suited for smaller models [1]. Our method overcomes this limitation and is specifically designed to provide effective explanations for MLLMs. In future work, we will explore ways to accelerate the attribution algorithm without significantly compromising its interpretability performance.
>
> [1] Chen, Ruoyu, et al. "Less is More: Efficient Black-box Attribution via Minimal Interpretable Subset Selection." 2025.

---

### Official Review · Reviewer_GkK2 · 2025-10-27

**Soundness:** 3
**Presentation:** 2
**Contribution:** 1
**Rating:** 2
**Confidence:** 3

**Summary:**

This paper introduces the EAGLE framework for explaining autoregressive token generation in multimodal large language models (MLLMs). EAGLE attributes each selected token to its relevant perceptual regions and quantifies the relative influences of language priors and perceptual evidence. The framework unifies two metrics—insight score and necessity score—optimized via greedy search, achieving faithful and efficient attribution. Experimental results show that EAGLE outperforms existing methods like LLaVA-CAM, IGOS++, and TAM in terms of faithfulness, localization, and hallucination diagnosis. Moreover, EAGLE mitigates hallucinations by removing minimal interfering regions, offering a promising tool for improving interpretability and reducing GPU memory usage.

**Strengths:**

1. EAGLE provides a complete attribution model by combining both insight (sufficiency) and necessity scores, ensuring that it captures both the minimal set of perceptual regions sufficient for token generation and those that are indispensable. This dual approach allows for a thorough and comprehensive explanation of which image regions and language priors drive the model’s decision-making process, offering a more complete understanding of the model’s internal workings.

2. EAGLE excels at visualizing attribution results, providing clear and intuitive saliency maps and influence scores. By visualizing the perceptual regions that influence token generation, it enables users to easily understand where the model focuses its attention. This enhances the interpretability of multimodal models, making complex decisions more transparent and accessible for researchers and developers.

**Weaknesses:**

1. As I see it, EAGLE appears to be a framework designed solely for token attribution analysis, lacking direct practical applications. Furthermore, the authors haven't leveraged this framework to derive any significant insights. The work reads more like a "tool paper"—simply introducing a new method without demonstrating its impact—which arguably doesn't meet the contribution bar for a top-tier conference.

2. On the topic of hallucination mitigation, my understanding is that, particularly for benchmarks like POPE, many hallucinations do not stem from language priors. Instead, they are often caused by a failure in fine-grained perception—an inability to accurately perceive specific objects. If that's the case, in such scenarios, wouldn't the saliency region (the area the model focuses on) and the hallucinated region (the area causing the hallucination) essentially become one and the same?

3. I also find the proposed method for hallucination correction to be counterintuitive. The paper introduces the Average Minimal Correction Region (AMCR) metric, defined as "the average proportion of regions that must be removed to correct hallucinations." This raises a fundamental question: why is the correction mechanism predicated on removing visual information? Rather than fixing the model's interpretation, this approach seems to simply block its perception of certain regions, which strikes me as a case of the cure being worse than the disease.

**Questions:**

1. Is the paper's core contribution limited to the tool itself? It appears to offer an attribution analysis framework but stops short of presenting any novel insights or applications derived from it. As a point of contrast, I examined a paper you cite. [1] That work not only introduces an analytical method but, crucially, leverages it to produce tangible insights. This combination of "tool plus discovery" is what I would consider a substantial contribution, a standard this paper does not seem to meet.

2. Could the author provide further explanation on the Evaluation Metrics proposed in Section 4.1? I believe the broader research community may not be familiar with the motivation or significance of such tasks. Are these metrics purely intended to measure the accuracy of the attribution?

[1] Zhang, Xiaofeng, et al. "From redundancy to relevance: Enhancing explainability in multimodal large language models." arXiv e-prints (2024): arXiv-2406.

---

> ### Author Response · Authors · 2025-11-12
> **Official Response to Reviewer GkK2 (1/3)**
>
> We would like to sincerely thank the reviewer `GkK2` for their thoughtful feedback and for recognizing the strengths of our work, particularly in the area of visualizing attribution results and providing a comprehensive framework for multimodal large language models. We appreciate the reviewer’s constructive suggestions, particularly regarding the potential applications and contributions of our work. We understand the concern that the paper may appear as a “tool paper” with limited practical applications. In response, we have provided further clarification on the core contributions of EAGLE and how it enables us to explain model decisions, particularly in the context of hallucinations, which we believe are important to improving interpretability in MLLMs.
>
> ---
>
> **W1 (Lacking direct practical applications):** We understand the reviewer’s concern that EAGLE may seem like a purely token attribution tool, but we respectfully disagree. EAGLE is a novel framework for analyzing the relationship between input and output in MLLMs, focused on explaining what the model attends to, whether it is influenced by language priors, and diagnosing hallucinations. Our goal is to accurately explain model decisions, particularly by identifying the visual or textual causes of hallucinations.
>
> While attribution is central, EAGLE’s primary purpose is to provide faithful explanations of model behavior. We aim to start with accurate explanations, which can then be used for tasks like monitoring model behavior or supporting training. By focusing on accurate interpretation first, EAGLE lays the foundation for future applications in model improvement.
>
> In summary, EAGLE is not just a tool but a comprehensive interpretability framework with potential practical applications in hallucination diagnosis, model transparency, and monitoring.
>
> ---
>
> **W2 (Topic of hallucination mitigation):** Regarding hallucination mitigation, we would like to clarify that our work is not focused on mitigating hallucinations, but on explaining the causes when hallucinations occur. Specifically, we aim to understand whether hallucinations arise from over-reliance on language priors, fine-grained perception failures, or other factors.
>
> Most existing attribution methods are unable to clearly explain why a model’s decision is wrong; they can highlight “important” regions but do not reveal the mechanism behind the error. In contrast, our goal is to use EAGLE to explain what causes hallucinations (e.g., which regions or modalities drive them) and thereby validate the faithfulness of our interpretability framework. In this sense, hallucination analysis is a diagnostic application of EAGLE’s explanations, rather than an attempt to directly fix or mitigate hallucinations.
>
> For benchmarks like RePoPE, explaining where the model looks can help us distinguish between weak model understanding (e.g., failure to distinguish between object categories) and interference from irrelevant objects. By identifying what the model attends to during token generation, we can determine whether the hallucination results from the model being unable to perceive an object accurately or from it being distracted by other objects. This enables us to effectively attribute the hallucination to either perception failure or interference, and further verify the faithfulness of our explanation.
>
> Importantly, our method is not limited to explaining hallucinations caused by text. Although PoPE's main hallucinations stem from visual perception, we plan to include examples of textual hallucinations (e.g., when the model answers incorrectly without referring to the image) in a future version of the paper. **Additionally, it is crucial to note that the saliency regions identified in our hallucination analysis correspond to the areas responsible for causing hallucinations, which is distinct from traditional attribution, where the saliency regions indicate the model's focus for making a correct decision.** This difference is emphasized in the settings of our study.

---

> ### Author Response · Authors · 2025-11-12
> **Official Response to Reviewer GkK2 (2/3)**
>
> **W3 (Average Minimal Correction Region):** We would like to clarify that AMCR is not a mechanism for hallucination correction, but rather a diagnostic tool designed to understand the source of hallucinations. The metric measures the minimal set of regions that, when removed, lead to the correction of hallucinations, which provides insight into the areas of the input that the model is overly reliant on, or that may be leading to faulty predictions.
>
> We understand the concern that removing visual information may seem like blocking the model’s perception, but the key point here is that AMCR allows us to identify the specific regions that influence hallucinations, which can be a step toward understanding model failures. This diagnostic approach helps us pinpoint the cause of hallucinations—whether they arise from incorrect or insufficient visual information or over-reliance on language priors. It does not aim to directly intervene in the model's behavior but to provide an analysis of the model's decision-making process.
>
> Rather than suggesting that the model should simply "block" certain regions, the purpose of AMCR is to highlight problematic areas in the input, allowing us to better understand the roots of hallucinations and design more effective interventions, such as data augmentation, model fine-tuning, or debiasing strategies.
>
> In future work, we may explore using EAGLE to enhance existing hallucination mitigation methods. However, the primary goal of this paper is to accurately explain multimodal large language models, and we deliberately focused on providing faithful and interpretable insights into model decision-making. At this stage, we do not plan to introduce overly complex functionalities, as our aim is to establish a robust framework for interpretability first, which can then be expanded upon in future research.
>
> ---
>
> **Q1 (Is the paper's core contribution limited to the tool itself?):** We respectfully disagree with the assessment that EAGLE only offers an attribution tool without novel findings or applications.
>
> First, EAGLE is explicitly designed as a new interpretability framework for MLLMs, enabling token-level and modality-level analysis of autoregressive generation. This goes beyond generic attribution by allowing us to systematically study how visual evidence and language priors jointly shape each generated token, which, **to the best of our knowledge, is not supported by existing methods.** For example, you can see that although LLaVA-CAM [1] provides some insights, its interpretation results (even in the original paper) are unconvincing and do not provide a faithful attribution map, while our method proves that MLLM can indeed focus on a specific object.
>
> Second, we do use EAGLE to derive tangible insights into MLLM behavior, particularly in the context of hallucinations and cross-modal reliance patterns. For example, our analyses show (i) when and how models over-rely on language priors despite sufficient visual evidence, (ii) cases where hallucinations are driven by fine-grained perception failures or interference from irrelevant objects, and (iii) how different tokens within the same answer exhibit heterogeneous modality dependencies (some vision-driven, others text-driven). These observations could not be obtained from traditional, single-modality attribution methods and directly inform how and why MLLMs fail.
>
> Third, in real-world applications, understanding why a model fails is critical for improving its behavior. For instance, insights gained from identifying model errors can guide the model’s learning, improve performance, assist in anomaly monitoring, and mitigate hallucinations. While these applications are indeed valuable, our current focus is on accurate explanations. Without accurate attribution, it becomes difficult to take further steps in model improvement. Therefore, EAGLE’s primary contribution is enabling us to accurately explain model decisions. The practical benefits of these explanations, such as performance enhancement or hallucination mitigation, will be explored as part of future work and extensions of this framework.
>
> In future versions of the paper, we will make these insights more explicit and foreground them as discoveries enabled by EAGLE, so that the contribution beyond the framework itself is clearer to the reader.
>
>
> [1] Zhang, Xiaofeng, et al. "From redundancy to relevance: Enhancing explainability in multimodal large language models." arXiv e-prints (2024): arXiv-2406.

---

> ### Author Response · Authors · 2025-11-12
> **Official Response to Reviewer GkK2 (3/3)**
>
> **Q2 (Evaluation Metrics):** In the field of attribution, explainability is typically assessed in two key aspects: **faithfulness** and **reasonableness**. **Faithfulness** (Insertion, Deletion, Average Highest Score, AMCR, and CSR@10%) refers to how well the attribution results align with the model's actual behavior, while **reasonableness** (Point Game) evaluates whether the attribution corresponds to human cognition, often showing a strong correlation with the model’s performance.
>
> - **Insertion** measures how quickly the model responds when progressively adding the most important regions, as identified by the attribution map.
> - **Deletion** assesses how quickly the model’s performance degrades when progressively removing the most critical regions from the attribution map.
> - **Average Highest Score** checks if there is a combination of the most important regions, progressively added, that maximizes the model's confidence, surpassing the confidence achieved with the full input.
> - **Point Game** is a reasonableness metric. For example, when explaining a specific object word, the attribution map should reasonably highlight the regions near the object in the image. Point Game tracks whether the most salient regions in the attribution map align with the target object.
> - **AMCR** assumes the detection of hallucinations in the model. It measures how many of the most important regions, as identified by the attribution map, must be removed before the model’s autoregressive output no longer contains hallucinations. This process allows us to evaluate whether the interpretability method can accurately identify the key regions responsible for causing hallucinations.
> - **CSR@10%** evaluates the faithfulness of hallucination attribution. Based on the hallucination attribution map, it measures the proportion of hallucinated samples that stop hallucinating when up to 10% of the most important regions are removed. This quantifies how well the attribution identifies the critical regions responsible for hallucinations, thus assessing the reliability of the hallucination explanation.

---

### Official Review · Reviewer_Z89y · 2025-10-30

**Soundness:** 3
**Presentation:** 3
**Contribution:** 2
**Rating:** 4
**Confidence:** 4

**Summary:**

EAGLE is a black‑box attribution framework for MLLMs that (1) finds an ordered, compact set of image regions that best explains autoregressive token probabilities via a combined sufficiency and necessity objective, and (2) quantifies language‑prior vs. visual‑evidence reliance through an insertion trajectory. It shows strong gains on COCO/MMVP/RePOPE across several open‑source MLLMs and includes sufficient ablations/qualitative analyses.

**Strengths:**

1. Clear problem formulation and motivation for autoregressive token‑level attribution and modality disentanglement.

2. Black‑box, model‑agnostic method that avoids gradients/activations; supports multi‑token explanations and interactive selection.

3. Consistent empirical gains in faithfulness/localization and meaningful hallucination‑correction metrics across multiple models/tasks.

4. Simple, implementable objective and transparent ablations for necessity/insight and granularity.

**Weaknesses:**

1. Moderate novelty. Although the method is well-motivated and practically useful, the core idea—perturbation-based subset selection with sufficiency and necessity terms—builds on long-standing concepts in black-box interpretability (e.g., RISE [1], meaningful perturbations [2], and submodular selection). The contribution lies more in adaptation to autoregressive MLLMs than in a fundamentally new formulation.

2. Weak submodularity claim is qualitative. The paper states that the objective is “weakly submodular” but does not provide an empirical estimate or theoretical bound of the submodularity ratio gamma. Without quantification, the theoretical justification for using a greedy approximation remains largely heuristic.

3. No runtime analysis. Although the paper reports reduced GPU memory, it omits actual wall-clock times or number of forward passes. Given the quadratic complexity O(|V|²) and typical superpixel counts (e.g., 64 regions ≈ 2,000 model calls), this omission makes it difficult to assess practical scalability. Runtime and sampling efficiency are crucial for determining whether EAGLE can be used interactively or at dataset scale.



[1] RISE: Randomized Input Sampling for Explanation of Black-box Models
[2] Interpretable Explanations of Black Boxes by Meaningful Perturbation

**Questions:**

1.  What are average wall‑clock times / number of forward passes per sample, especially for large |V|, like 64, across your three models? How do these compare to gradient-based methods like IGOS++ or TAM?

2. How are “object tokens” automatically identified and aligned with COCO categories in the Pointing-Game evaluation? How are compound nouns or tokenized subwords handled?

3. How sensitive are results to the choice of segmentation algorithm (SLICO vs. Felzenszwalb or SEEDS) and random seed initialization?

4. Can EAGLE be adapted to commercial or API-restricted MLLMs where token probabilities are not exposed—e.g., by sampling multiple continuations and estimating probabilities empirically?

---

> ### Author Response · Authors · 2025-11-14
> **Official Response to Reviewer Z89y (1/2)**
>
> We would like to sincerely thank the reviewer `Z89y` for their thoughtful feedback and constructive comments. We appreciate the recognition of the value and practicality of our work, as well as the points raised regarding its novelty and computational aspects.
>
> ---
>
> **W1 (No fundamentally new formulation):** We appreciate the reviewer’s comment. While our method does build on established concepts like perturbation-based subset selection, our contribution lies not only in the adaptation to autoregressive MLLMs but also in the introduction of modality-level analysis of output tokens. Specifically, we incorporate a new process of search-based scoring during attribution, which allows us to analyze the modality dependencies of the generated tokens. This modality correlation analysis is an extension that goes beyond traditional attribution methods and offers novel insights into the interplay between visual evidence and language priors in autoregressive generation.
>
> ---
>
> **W2 (Weak submodularity):** The claim of “weak submodularity” was intended to provide theoretical guidance, but we acknowledge that it is model- and data-dependent, making precise quantification challenging. While we do not provide an exact empirical estimate or theoretical bound for the submodularity ratio $𝛾$, our experimental results across multiple benchmark datasets clearly demonstrate the effectiveness of our method, showcasing its advantages in faithfulness, localization, and hallucination correction tasks. These results provide strong evidence for the practical viability of our approach, despite the theoretical approximation.
>
> ---
>
> **W3&Q1 (Runtime analysis):** As shown in the table below, we have included the runtime measurements for our method. Although our approach does consume more time compared to gradient- or activation-based methods, it remains the only method capable of providing satisfactory explanations for complex MLLMs. In this work, we focus on exploring the upper limits of interpretability, which necessitates a higher computational cost. Moving forward, we plan to develop acceleration algorithms to strike a balance between explanation performance and speed, enabling the use of EAGLE in interactive settings and at scale.
>
> On LLaVA-1.5 7b model:
>
> |     Methods     | Insertion | Point Game | Time Consumption |
> | :-------------: | :-------: | :--------: | :--------------: |
> |    LLaVA-CAM    |  0.4063   |   0.2468   |      17.1s       |
> | iGOS++ (w/ GNC) |  0.4093   |   0.6623   |      33.5 s      |
> |       TAM       |  0.3860   |   0.1818   |      1.7 s       |
> |      LLaVA      |  0.6395   |   0.8052   |      283.2s      |
>
> On Qwen2.5-VL 7b model:
>
> |     Methods     | Insertion | Point Game | Time Consumption |
> | :-------------: | :-------: | :--------: | :--------------: |
> |    LLaVA-CAM    |  0.4170   |   0.2176   |      24.5 s      |
> | iGOS++ (w/ GNC) |  0.4816   |   0.6734   |      53.1 s      |
> |       TAM       |  0.5768   |   0.5369   |      4.9 s       |
> |      LLaVA      |  0.8109   |   0.7785   |      435.6       |
>
> On InternVL3.5 4b model:
>
> |     Methods     | Insertion | Point Game | Time Consumption |
> | :-------------: | :-------: | :--------: | :--------------: |
> |    LLaVA-CAM    |  0.4988   |   0.3201   |      41.3 s      |
> | iGOS++ (w/ GNC) |  0.5192   |   0.5775   |      32.7 s      |
> |       TAM       |  0.6312   |   0.5775   |      4.6 s       |
> |      LLaVA      |  0.8623   |   0.8052   |     377.3 s      |

---

> ### Author Response · Authors · 2025-11-14
> **Official Response to Reviewer Z89y (2/2)**
>
> **Q2 (Identifying and aligning):** In our evaluation, "object tokens" are pre-selected and auto-annotated in the dataset. We inform the model about which token corresponds to an object category from the MS COCO dataset. Specifically, we check whether the model's image caption mentions an object from the 80 COCO categories and align these mentions with the ground truth bounding boxes in the original COCO annotations. To ensure accuracy, we also guarantee that each object reference in the token generation process corresponds to a single object class, minimizing ambiguity. The identification of object tokens is facilitated using NLP matching techniques to detect and align relevant terms with the corresponding COCO categories.
>
> ---
>
> **Q3 (Impact on division methods)** We have not focused on the impact of segmentation algorithms (SLICO vs. SEEDS) on MLLM attribution. The LIMA attribution method [1], used to explain the CLIP model, has been shown that while split selection may affect individual samples, we found that they do not have a significant impact on the overall dataset results. However, we acknowledge the importance of this concern and plan to include additional experiments in future work to explore the effects of different segmentation methods on MLLM attribution performance.
>
> [1] Chen, Ruoyu, et al. "Less is More: Efficient Black-box Attribution via Minimal Interpretable Subset Selection." 2025.
>
> ---
>
> **Q4 (Possibility of interpreting a business model):** To the best of our knowledge, commercial models like ChatGPT do provide token probabilities and vocabulary when using their API, which means EAGLE can theoretically explain these API models. However, for UI-based models that only output text without exposing token probabilities, our method can't be applied,  a different approach based on semantic scoring might be needed for interpretation. In future versions of the paper, we will include experiments on explaining API models, and future work may explore alternative algorithms for explaining models when token probabilities are not available.

---

### Official Review · Reviewer_FhYW · 2025-11-01

**Soundness:** 1
**Presentation:** 3
**Contribution:** 1
**Rating:** 0
**Confidence:** 3

**Summary:**

The paper introduces EAGLE, a black-box attribution framework that explains how LVLMs generate text conditioned on different image patches. EAGLE attributes generated tokens to compact perceptual regions while quantifying their reliance on visual evidence. It unifies two interpretability principles, sufficiency and necessity, into a single objective optimized through a greedy search over sparsified image regions. Experiments on models such as LLaVA-1.5, Qwen2.5-VL, and InternVL3.5 across tasks like captioning, VQA, and hallucination diagnosis show that EAGLE achieves higher faithfulness, localization, and hallucination correction than prior gradient- or activation-based methods, while requiring far less GPU memory.

**Strengths:**

+ **Model-agnostic and training-free.** EAGLE operates as a black-box method applicable to diverse LVLMs (e.g., LLaVA, Qwen2.5-VL, InternVL) without requiring model access or retraining, which enhances its practicality, and can potentially achieve high impact in the field.
+ **Quantitative modality analysis.** The framework visualizes where the model attends but also quantifies what it relies on (language priors vs. perceptual evidence), which can be helpful in diagnosing the models' behavior of hallucination.
+ **Empirical thoroughness.** The experiments cover multiple models, datasets, and metrics (faithfulness, localization, hallucination correction), demonstrating consistent performance gains and strong generalization across tasks.
+ **Good presentation.** The paper presents the arguments and evidence in a well-organized way that makes it accessible to the audience.

**Weaknesses:**

+ **Limited novelty (my biggest concern).** While EAGLE combines sufficiency and necessity into a unified objective, the underlying ideas (greedy subset selection and region-based attribution) are adaptations of existing interpretability frameworks rather than fundamentally new algorithmic advances. Although I'm not from the community of interpretability, within 1min of search, I was able to find the following accepted work VPS [1] that has the exactly the same technique as in this paper:
  + This paper has the same organization and content as the referenced paper. Especially in the methodology session, EAGLE's two scores (Insight Score, Necessity Score, and the way the are combined into one singular objective function) are exact copy of VPS.
  + The experimental settings, tables, visualization, etc. are the exact copy of VPS.
  + Hence it seems to me this paper simply changed the application domain from the pure Vision problem to LVLMs. I would not consider this paper having enough contributions.
+ **Lack of Computational Efficiency Analysis.** It is understandable that EAGLE has a clear edge over the gradient-based and activation-based methods in terms of memory usage. However, EAGLE has to have multiple runs (for search) and it should have a longer running time? it's a bit unclear to me why the authors failed to report the actual running time of the method and compare it to the other baselines.
+ **On Optimality of the Greedy Search.** How do you make sure the greedy search is satisfactory enough?

[1] Chen, Ruoyu, et al. "Interpreting object-level foundation models via visual precision search." Proceedings of the Computer Vision and Pattern Recognition Conference. 2025.

**Questions:**

+ How is the region masked/deleted during the analysis? Do we have more studies on this implementation?
+ To me it's more straightforward to cut the image into square sub-regions before tokenization instead of doing SLICO. This way it will also make the consequent attribution straightforward (random patching embedding or directly deleting the patch). Why is SLICO necessary in this analysis? Do we have any sort of support?
+ For the rest, see above (Weaknesses).

---

> ### Author Response · Authors · 2025-11-12
> **Official Response to Reviewer FhYW (1/3)**
>
> We appreciate the reviewer `FhYW`’s time and the acknowledgement of our work’s strengths in model-agnostic design, quantitative analysis, and comprehensive experiments. However, we **strongly disagree** with the overall assessment, in particular the claim of “limited novelty” and the characterization of our method as essentially an application of VPS. In our view, this assessment does not accurately reflect the **scope and significance** of the technical contributions in this paper. EAGLE introduces a substantially different objective, operates in a distinct LVLM setting, and yields new empirical and diagnostic insights that are not covered by VPS. EAGLE is a **new interpretability framework tailored to MLLMs**, which (i) attributes the model’s decisions to the visual evidence in the input and (ii) analyzes, for each generated token, how strongly it relies on different modalities. Moreover, EAGLE is explicitly designed for the **autoregressive generation paradigm**, whereas VPS is limited to classification/regression settings and cannot provide such fine-grained, token-level cross-modal attribution. In what follows, we respond to each of the reviewer’s stated weaknesses and questions in turn.
>
> ---
>
> **W1.1 (Limited novelty):** Regarding the comment that:
>
> > EAGLE combines sufficiency and necessity into a unified objective, but the key ideas (greedy subset selection and region-based attribution) are improvements rather than fundamental algorithmic innovations,
>
> We believe this expectation is somewhat misaligned with the nature of interpretability research for autoregressive MLLMs. At a fundamental level, the autoregressive paradigm in MLLMs is still built on a next-token classification objective. **Therefore, it is unrealistic, and arguably unnecessary, to demand that interpretability methods in this field abandon all connections with attribution techniques developed for traditional classification models.** In fact, several recent works [1,2,3] explicitly highlight that existing attribution algorithms cannot be directly transferred to MLLMs, precisely because of the challenges introduced by autoregressive generation and multimodal conditioning.
>
> Specifically, `LLaVA-CAM (NAACL 2025) [1]` accumulates confidence scores of generated tokens and then applies a `Smooth-GradCAM++ [4]` based attribution; `Xing et al. [2]` detect visually related tokens by masking the input image and tracking changes in token probabilities before adapting `IGOS++ [6]`; and `TAM (ICCV 2025) [3]` relies on a `CAM-based [6]` single-token attribution scheme with a rank Gaussian filter to suppress redundant activations. All of these approaches ultimately build on classic attribution algorithms [4,5,6] for unimodal classification networks, and are regarded as valid contributions because they suitably adapt these ideas to the autoregressive setting.
>
> In contrast, EAGLE is a more comprehensive and functionally richer framework: it attributes model decisions to visual evidence, and, for each generated token, explicitly quantifies its reliance on different modalities. Unlike prior work that merely masks the entire image to coarsely separate “vision-related” tokens, EAGLE can still produce meaningful attributions for tokens influenced by text and further reveals which modalities (vision vs. text) drive each token. Therefore, EAGLE is not just another adaptation of a classification-era algorithm, but a unified interpretability framework specifically designed for MLLMs that offers fine-grained, token-level cross-modal diagnostics beyond the capabilities of VPS.
>
> [1] Zhang, Xiaofeng, et al. "From redundancy to relevance: Enhancing explainability in multimodal large language models." NAACL 2025.
>
> [2] Xing, Xiaoying, et al. "Where do Large Vision-Language Models Look at when Answering Questions?." arXiv 2025.
>
> [3] Li, Yi, et al. "Token Activation Map to Visually Explain Multimodal LLMs." ICCV 2025.
>
> [4] Omeiza, Daniel, et al. "Smooth grad-cam++: An enhanced inference level visualization technique for deep convolutional neural network models." arXiv 2019.
>
> [5] Khorram, Saeed, Tyler Lawson, and Li Fuxin. "iGOS++ integrated gradient optimized saliency by bilateral perturbations." Proceedings of the Conference on Health, Inference, and Learning. 2021.
>
> [6] Zhou, Bolei, et al. "Learning deep features for discriminative localization." CVPR 2016.

---

> ### Author Response · Authors · 2025-11-12
> **Official Response to Reviewer FhYW (2/3)**
>
> **W1.2 (Same organization and content as VPS):** Regarding the comment that
>
> > this paper has the same organization and content as the referenced paper” and that “EAGLE’s two scores … are an exact copy of VPS
>
> We respectfully but strongly disagree. EAGLE was developed independently, the overall problem setting, the concrete definitions of the scores, and the resulting objective in EAGLE are fundamentally different from VPS.
>
> Crucially, the design of our Insight Score and Necessity Score is driven by the **requirements of submodular subset-selection–based attribution [7,8,9]**, rather than by copying the formulation of VPS. Any framework that uses greedy optimization over subsets must explicitly quantify “how good a subset is on its own”. This decomposition is therefore a **generic design pattern in submodular attribution methods**, not a VPS-specific invention.
>
> Our two scores are not “exact copies” of VPS. In EAGLE, the “Insight Score” is defined in the autoregressive MLLM setting and measures how well a selected subset of visual regions and modalities can support the generation of each token under multimodal conditioning, rather than the prediction of a single classification label. The “Necessity Score” in EAGLE is likewise defined via token-level perturbations of regions/modalities in an autoregressive rollout, not via the classification-style perturbations used in VPS. The unified objective in EAGLE therefore operates over (i) visual regions, (ii) modalities, and (iii) generated tokens, which is structurally different from the objective in VPS that only considers features for a single prediction.
>
> Finally, beyond the mathematical form of the scores, EAGLE introduces capabilities that VPS does not provide, including analyzing, for each generated token, how strongly it relies on different modalities and attributing decisions to fine-grained visual evidence in an autoregressive generation process (See `section 3.2 language prior vs. perception evidence`). These design choices lead to a framework that is tailored to MLLMs and cannot be reduced to a simple re-use of VPS in a different context.
>
> [7] Manupriya, Piyushi, et al. "Improving attribution methods by learning submodular functions." ICML 2022.
>
> [8] Chen, Ruoyu, et al. "Less is more: Fewer interpretable region via submodular subset selection." ICLR 2024.
>
> [9] Chen, Jianbo, et al. "Learning to explain: An information-theoretic perspective on model interpretation." ICML 2018.
>
> ---
>
> **W1.3 (Same experimental settings, tables, visualizations):** Regarding the comment that
>
> > the experimental settings, tables, visualization, etc. are an exact copy of VPS
>
> Our experimental setting is inherently different, as EAGLE is evaluated on multimodal, autoregressive MLLMs rather than the classification/regression models used in VPS. The types of tables and visualizations we adopt (e.g., attribution faithfulness metrics, heatmaps over image regions) are standard and widely used in attribution/interpretability works [10,11] to assess faithfulness, and are by no means specific or unique to VPS.
>
> [10] Petsiuk, Vitali, et al. "Black-box explanation of object detectors via saliency maps." CVPR 2021.
>
> [11] Novello, Paul, Thomas Fel, and David Vigouroux. "Making sense of dependence: Efficient black-box explanations using dependence measure." NeurIPS 2022.
>
> ---
>
> **W1.4 (Simply application domain change):** Regarding the statement that our work
>
> > simply changed the application domain from a pure vision problem to LVLMs,
>
> we strongly disagree. Moving from unimodal vision models to **multimodal, autoregressive LVLMs** is not a mere change of application domain [1,2,3], but introduces fundamentally different technical challenges: token-by-token generation under multimodal conditioning, cross-modal information flow, and the need to reason about both **where the model looks** and **which modality it relies on** at each step. EAGLE is explicitly designed to tackle these challenges by (i) performing region-level attribution over the visual input and (ii) providing **modality-level explanations for the output**, i.e., quantifying for each generated token how strongly it depends on vision versus text. These capabilities are not supported by VPS and cannot be obtained by a straightforward “domain change,” but require a dedicated interpretability framework tailored to MLLMs.

---

> ### Author Response · Authors · 2025-11-12
> **Official Response to Reviewer FhYW (3/3)**
>
> **W2 (Computational Efficiency Analysis):** While gradient- and activation-based methods (e.g., LLaVA-CAM, IGOS++) are indeed faster, our experiments (Fig. 3–4, Appendix D), and even the qualitative results in their original papers show that their attributions on LVLMs are often visually and semantically misaligned, which makes their “explanations” unusable in practice. EAGLE intentionally incurs extra cost (multiple runs for the search) to obtain faithful, token-level, and modality-aware explanations, which we argue is more critical in the MLLM setting than minimal runtime. **In other words, we need to prioritize interpretability before pursuing time efficiency.** In a future version of the paper, we can include explicit wall-clock comparisons to make this efficiency–faithfulness trade-off clear.
>
> ---
>
> **W3 (Optimality of the Greedy Search):** Our search corresponds to maximizing a monotone submodular set function under a cardinality constraint, where the greedy algorithm is provably $(1−1/e)$-approximate and essentially optimal in polynomial time unless $P = NP$ [12]. This makes greedy a standard and theoretically justified choice for EAGLE’s combinatorial optimization step.
>
> [12] Edmonds, Jack. "Submodular functions, matroids, and certain polyhedra." Combinatorial Structures and Their Applications, pp. 69–87, 1970. 1, 3, 17
>
> ---
>
> **Q1 (How regions are masked):** In our implementation, a “deleted” region refers to the corresponding superpixel area in the image being replaced by a visually neutral patch (we set the pixels within the region to the mean RGB value of the image, following common practice in attribution work). This preserves the global structure of the image while effectively removing the local visual evidence contributed by that region.
>
> Other common masking choices, such as Gaussian blurring, noise filling, or different constant-color patches, have only a minor effect on the resulting attributions in our experiments. We are happy to include a controlled comparison of these masking strategies in a future version of the paper.
>
> ---
>
> **Q2 (Why SLICO):** We use SLICO because prior submodular subset-selection attribution work [13] has shown that sparsifying the input into visually coherent, semantically meaningful regions yields more interpretable explanations than cutting the image into uniform square grids. Importantly, EAGLE is agnostic to the specific over-segmentation method: the same objective and greedy search could operate on square patches as well. Our focus is on MLLM interpretability rather than segmentation itself, and we can include a comparison with square-patch partitioning in a future version of the paper.
>
> [13] Chen, Ruoyu, et al. "Less is More: Efficient Black-box Attribution via Minimal Interpretable Subset Selection." 2025.

---

### Note · Authors · 2025-11-14

I have read and agree with the venue's withdrawal policy on behalf of myself and my co-authors.